# Taphonomic and Diagenetic Pathways to Protein Preservation, Part II: The Case of *Brachylophosaurus canadensis* Specimen MOR 2598

**DOI:** 10.3390/biology11081177

**Published:** 2022-08-05

**Authors:** Paul V. Ullmann, Richard D. Ash, John B. Scannella

**Affiliations:** 1Department of Geology, Rowan University, Glassboro, NJ 08028, USA; 2Department of Geology, University of Maryland, College Park, MD 20742, USA; 3Museum of the Rockies, Montana State University, Bozeman, MT 59717, USA; 4Department of Earth Sciences, Montana State University, Bozeman, MT 59717, USA

**Keywords:** REE, *Brachylophosaurus*, molecular paleontology, geochemical taphonomy, diagenesis, bone, protein, collagen, Judith River Formation

## Abstract

**Simple Summary:**

Reports of the recovery of proteins and other molecules from fossils have become so common over the last two decades that some paleontologists now focus almost entirely on studying how biologic molecules can persist in fossils. In this study, we explored the fossilization history of a specimen of the hadrosaurid dinosaur *Brachylophosaurus* which was previously shown to preserve original cells, tissues, and structural proteins. Trace element analyses of the tibia of this specimen revealed that after its bones were buried in a brackish estuarine channel, they fossilized under wet conditions which shifted in redox state multiple times. The successful recovery of proteins from this specimen, despite this complex history of chemical alterations, shows that the processes which bind and stabilize biologic molecules shortly after death provide them remarkable physical and chemical resiliency. By uniting our results with those of similar studies on other dinosaur fossils known to also preserve original proteins, we also conclude that exposure to oxidizing conditions in the initial ~48 h postmortem likely promotes molecular stabilization reactions, and the retention of early-diagenetic trace element signatures may be a useful proxy for molecular recovery potential.

**Abstract:**

Recent recoveries of peptide sequences from two Cretaceous dinosaur bones require paleontologists to rethink traditional notions about how fossilization occurs. As part of this shifting paradigm, several research groups have recently begun attempting to characterize biomolecular decay and stabilization pathways in diverse paleoenvironmental and diagenetic settings. To advance these efforts, we assessed the taphonomic and geochemical history of *Brachylophosaurus canadensis* specimen MOR 2598, the left femur of which was previously found to retain endogenous cells, tissues, and structural proteins. Combined stratigraphic and trace element data show that after brief fluvial transport, this articulated hind limb was buried in a sandy, likely-brackish, estuarine channel. During early diagenesis, percolating groundwaters stagnated within the bones, forming reducing internal microenvironments. Recent exposure and weathering also caused the surficial leaching of trace elements from the specimen. Despite these shifting redox regimes, proteins within the bones were able to survive through diagenesis, attesting to their remarkable resiliency over geologic time. Synthesizing our findings with other recent studies reveals that oxidizing conditions in the initial ~48 h postmortem likely promote molecular stabilization reactions and that the retention of early-diagenetic trace element signatures may be a useful proxy for molecular recovery potential.

## 1. Introduction

### 1.1. Shifting Views on Molecular Preservation

Fossilization has historically been viewed as a “harsh” process involving forced mineralization coincident with the wholesale loss of organic tissues and their component biomolecules (e.g., [1]). However, an ever-expanding wealth of recent studies employing methods from histochemistry and immunoassays to genomics and proteomics have shattered this ‘traditional’ paradigm. As reviewed by ourselves and others [2,3,4,5,6,7], it is now clear that not only is the long-term preservation of endogenous DNA, proteins, and other biomolecules possible in fossils, but select burial circumstances may actually promote molecular preservation in both plants and animals interred in a diverse array of depositional environments. For example, drastic advances in analytical resolution over the last two decades have enabled the recovery of nuclear and mitochondrial genomes and/or proteomes from fossils of a number of Pleistocene mammals, such as cave bears [8], mammoths [9,10,11,12], horses [13], saber-toothed cats [14], and ancient hominins [15,16].

Given their greater resilience to decay than DNA [17], structural proteins (e.g., collagen I, actin, and β-keratin) are now known from vertebrate fossils dating all the way back to the Jurassic (e.g., [18,19,20,21,22]). Considerable attention has been paid especially to the protein collagen I due to its sheer abundance in bioapatitic tissues [23] and inferred high preservation potential [18,24,25,26,27,28,29,30,31,32,33,34,35,36,37,38,39,40,41,42,43,44,45,46,47]. Recent studies have employed numerous independent techniques to identify collagen within fossils, with arguably the most convincing evidence being the identification of original peptide sequences and diagenetiforms (protein remains demonstrably modified by diagenetic alterations [48]) in fossil bones via high-resolution tandem mass spectrometry [18,28,30,42,45]. Remarkably, cladistic analyses incorporating collagen I peptides recovered from two Cretaceous nonavian dinosaur bones confirmed their archosaurian identities and thus endogeneity [45,49]. Such findings unequivocally demonstrate both that biomolecules can ‘survive’ fossilization and that portions of them can persist over strikingly-long geologic timescales.

However, the idea of molecular preservation in Mesozoic (and possibly even older) fossils remains controversial to some due to our incomplete understanding of soft-tissue fossilization in general and the geochemical reactions which may stabilize biomolecules within cells and tissues over such immense time frames. While it is universally agreed that processes such as rapid burial can facilitate the preservation of soft tissues in fossils [50,51], it largely remains unclear how other taphonomic processes and the physical (e.g., sedimentology and hydrodynamics) and chemical attributes (e.g., aqueous geochemistry) of depositional environments influence decay at the molecular level (but see [52] for an informative initial foray into this subject). It is therefore vital for researchers to not only demonstrate the authenticity of biomolecular remnants in fossils but to also identify the physicochemical factors acting within the diagenetic settings which permitted such cases of “exceptional” preservation. Pioneering actualistic studies by Schweitzer et al. [4] and Boatman et al. [53] demonstrated that iron free radicals in diagenetic pore fluids likely play a role by inducing intra-molecular crosslinking, but are other aspects of groundwater chemistry (i.e., redox state; cf. [54]) and diffusion history (i.e., duration spent saturated; cf. [47]) equally important in determining whether or not biomolecules persist in fossils? Additionally, if they are, which depositional settings and diagenetic histories most favor long-term molecular preservation? In short, we are just beginning to explore these questions.

### 1.2. Insights from Trace Element Analyses

One of the most effective means of clarifying the geochemical history of a fossil is through studying its trace element composition. After being solubilized from surrounding sediments by percolating groundwaters via oxidation, dissolution, and other processes, trace element ions, including those of the rare earth elements (REE: lanthanum–lutetium), uranium, and scandium, are ubiquitously adsorbed by bone hydroxyapatite during diagenesis [55]. Since these elements are essentially absent in bone tissue in vivo, their presence in fossil bones derives almost entirely from postmortem interactions with surface and groundwaters [55,56]. As a result, the proportions and spatial distributions of trace elements within a fossil bone provide detailed insights into the chemistries of past pore fluids and the geochemical milieus to which a specimen was exposed throughout its term of burial (e.g., [57,58,59,60,61,62,63,64]). Trace element signatures have thus been successfully utilized to: (1) infer the relative degree of chemical alteration a specimen has endured [65,66]; (2) characterize the chemistry of pore fluids in past environments (e.g., [60,63,67]); (3) track spatiotemporal trends in redox conditions within specimens throughout diagenesis (e.g., [62,63,64]); and (4) clarify the number and relative timing of exposures to pore fluids throughout diagenesis (e.g., [64,68]). REE signatures, in particular, have also been shown to potentially be viable proxies for molecular preservation in fossil bones (i.e., they can help identify the most ideal specimens for paleomolecular investigation [47]).

Several of our recent studies capitalized on these diverse utilities of trace elements to elucidate the paleoenvironmental, geochemical, and diagenetic history of two fossils from the Cretaceous Hell Creek Formation which were each documented to preserve endogenous collagen I [28,47], specifically an *Edmontosaurus* fibula [63] and a femur of *Tyrannosaurus rex* specimen Museum of the Rockies (MOR) 1125 [64] (also see [69] for alternative taxonomic assignment of MOR 1125). These studies provided intriguing insights into taphonomic pathways to protein preservation, but they still merely constitute two case studies in the same geologic formation; the full suite of taphonomic and diagenetic variables at play in molecular preservation remains to be clarified.

In this study, we conducted trace element analyses on the only Mesozoic fossil other than *T. rex* MOR 1125 known to yield endogenous peptide sequences: *Brachylophosaurus canadensis* specimen MOR 2598. Schweitzer et al. [30] and Schroeter et al. [45] each recovered numerous peptides of collagen I from the left femur of this hadrosaur, the authenticity of which were independently corroborated by multiple forms of microscopy, infrared spectroscopy, mass spectrometry, and immunoassays replicated in multiple laboratories by separate researchers each using dedicated equipment and reagents (see each reference for further details). The cumulatively-comprehensive approach undertaken by these studies to demonstrate reproducibility and authenticate the endogeneity of collagen in MOR 2598 set a rigorous standard that has yet to be matched again, despite over a decade of ensuing research on other specimens. Given this great significance of MOR 2598 in providing concrete foundations for the field of molecular paleontology, it is only right to resolve the taphonomic and diagenetic history of this specimen in equally comprehensive detail.

## 2. Taphonomic and Geologic Context

MOR 2598 consists of an articulated left hind limb of a subadult *Brachylophosaurus canadensis* recovered from an outcrop of the Campanian Judith River Formation north of Malta, Montana, on lands managed by the Montana Department of Natural Resources and Conservation (Figure 1). The specimen was found within a thick sequence (~7 m) of trough cross-stratified channel sandstones exposed along the southern side of Cottonwood Creek. Schweitzer et al. [31] concluded that these strata were deposited in a fluvial channel within the overall lowland fluviodeltaic system of the Judith River ecosystems [70,71]. The tibia, fibula, and pes were collected in the summer of 2006, and the femur was collected in a separate plaster jacket the following year. The articulation of these skeletal elements implies they were still joined by connective tissues (i.e., ligaments) at the time of burial. The incomplete nature of the tibia (see below) and slightly lighter color of this bone than the femur indicate that portions of the tibia were exposed by modern erosion/weathering upon discovery (whereas the femur was not [30]). All of the skeletal elements are brown in color (e.g., Figure 1B), indicating their mineralogy has likely been transformed from hydroxyapatite to fluorapatite, which is typical of bone fossilization [72,73]. As discussed by Schweitzer et al. [30], the femur was collected with 10–12 cm of sediment still encasing it to maintain geochemical equilibrium for as long as possible before examination via demineralization, scanning electron microscopy, multiple immunoassays, and liquid chromatography–tandem mass spectrometry. All of the skeletal elements appear well-preserved morphologically in that they lack any signs of weathering or abrasion, but the tibia (examined herein) is missing its distal end and a section of the posterior and medial portions of the shaft near its proximal end. The tibia is also highly fractured, exhibiting numerous transverse and longitudinal fractures arising from compaction after fossilization. Though the medullary cavity of this bone is partly ‘filled’ due to compaction and the partial crushing of cancellous trabeculae, there are no signs of permineralization or infilling by the sedimentary matrix (pers. observations, and [74]).

## 3. Materials and Methods

### 3.1. Materials

It was not possible to acquire a sample from the left femur of *Brachylophosaurus canadensis* MOR 2598 for this study as prior histologic and paleomolecular studies of this particular skeletal element of the specimen [30,45,74] and reconstructive efforts undertaken during preparation to maintain permanent stability of the bone left no portion of the cortex easily removable without compromising the integrity of the fossil. Therefore, we instead extracted a fragment of the cortex from the midshaft of the left tibia of MOR 2598. This bone was found in articulation with the left femur in the field and was accordingly buried at the same stratigraphic position within the same stratum as the left femur, so it was almost certainly exposed to the same environmental conditions postmortem and early-diagenetic regime(s) after burial as the left femur. The cortex at the midshaft of tibiae also possesses a similar thickness, density, and histologic microstructure to the midshaft cortex of femora in hadrosaurids (e.g., [76,77]). For these reasons, we are confident that trace element signatures within the left tibia should be very similar to those that would be identified in the left femur examined by prior studies, and that this tibia therefore represents a suitable choice for examining the geochemical history of the hind limb of MOR 2598 as a whole.

The excised cortical fragment encompasses the majority of the cortical thickness of the bone, including the external margin, but fragile cancellous bone within the medullary cavity disintegrated away during preparation. Because of this, we infer that the innermost portion of the internal cortex was not included in our analyses.

### 3.2. Methods

#### 3.2.1. Sample Preparation

The cortical sample was embedded in Silmar 41^TM^ resin (US Composites, West Palm Beach, FL, USA) under vacuum, then sectioned using a Hillquist SF-8 trim saw (Hillquist, Arvada, CO, USA). The resulting ~3 mm-thick section was briefly rinsed with distilled water then briefly polished with 600 grit silicon carbide to acquire an evenly smooth surface for ensuing laser ablation–inductively coupled plasma mass spectrometry (LA-ICPMS) analyses.

#### 3.2.2. LA-ICPMS Analyses

We employed the same LA-ICPMS methods as Ullmann et al. [63,64] in this study and refer the reader to the Appendix A and those publications for thorough details. In brief, LA-ICPMS was chosen as a powerful means of examining the spatial distribution of REEs and other minor and trace elements within the tibia of MOR 2598, which can provide unique insights into the diagenetic history of a fossil bone and any geochemical shifts it endured through its fossilization. Concentrations of elements are reported in parts per million (ppm) except for iron, which is reported in weight percent (wt. %). REE concentrations were normalized against the North American Shale Composite (NASC; [78,79]) to facilitate comparisons to other fossil bones from other localities. The use of a subscript _N_ denotes NASC-normalized values and ratios. The reproducibility of our results was taken as the percent relative standard deviation for all REEs in the NIST 610 glass standard; it averaged 1.5% and was below 3% for all analyzed elements. NASC-normalized REE ratios were used to calculate (Ce/Ce*)_N_, (Ce/Ce**)_N_, (Pr/Pr*)_N_, and (La/La*)_N_ anomalies following Herwartz et al. [60]: (Ce/Ce*)_N_ = Ce_N_/(0.5*La_N_ + 0.5*Pr_N_), (Ce/Ce**)_N_ = Ce_N_/(2*Pr_N_−Nd_N_), (Pr/Pr*)_N_ = Pr_N_/(0.5*Ce_N_ + 0.5*Nd_N_), and (La/La*)_N_ = La_N_/(3*Pr_N_ − 2*Nd_N_).

## 4. Results

### 4.1. Overall REE Composition

As a whole (i.e., by summing all transect data), the left tibia of MOR 2598 exhibits a ∑REE value of 256 ppm (Table 1). The three most abundant trace elements in the cortex of the bone are iron (Fe), strontium (Sr), and barium (Ba), which exhibit concentrations of 0.94 wt. %, 2499 ppm, and 1448 ppm, respectively (Table 1). All of these elements, as well as manganese (Mn), are present in concentrations approximately one to two orders of magnitude higher than REEs (Table 1). Whereas the average scandium (Sc) enrichment (59 ppm) is around the same magnitude as those of most LREEs (~10–90 ppm), the average yttrium (Y) concentration (190 ppm) is more than double that of the highest REE (89 ppm for cerium, Ce; Table 1). Among REEs, there is substantially greater enrichment in light rare earth elements (LREEs, La–Nd) than middle (MREEs, Sm–Gd) and heavy rare earths (HREEs, Tb–Lu), clearly indicative of fractionation during uptake (see Discussion). The average whole-bone concentration of uranium (U), 51 ppm, is distinctly higher than those of bones from the Hell Creek Formation known to also yield endogenous collagen I (2–38 ppm [63,64]).

### 4.2. Intra-Bone Concentration Depth Profiles

Each REE exhibits a steeply-declining concentration profile from the cortical margin. As an example, lanthanum (La) concentrations decrease from ~1000 ppm at the outer edge of the cortex to ~10 ppm at 5 mm, thus constituting an order of magnitude decrease across this distance (Figure 2A). Concentrations of HREEs and the latter half of the MREE series, as well as U, Sc, Y, and lutetium (Lu), all increase toward the internal end of the transect (Figure 2A–C). For example, Yb concentrations increase from ~1–2 ppm at a depth of 25 mm to ~15–20 ppm at the internal end of the transect (Figure 2A, Appendix A). Such increases signify secondary diffusion from within the medullary cavity (see the Discussion and Appendix A). Among REEs, Ce exhibits the highest concentration at the cortical margin (~3000 ppm), whereas Lu exhibits the lowest (~10 ppm). LREEs generally exhibit the steepest concentration profiles, reflective of spatially-heterogeneous uptake, whereas HREEs generally exhibit flatter profiles, reflective of comparatively more spatially homogenous uptake. MREE profiles are generally intermediate in steepness, and MREE concentrations commonly fall below the lower detection limit in the middle and internal cortices (Appendix A).

Brief spikes in concentrations typically encountered in osteonal tissue around Haversian canals are rare and generally of miniscule magnitude. Although this would seem to imply a lack of major uptake through vascular systems, most REE (e.g., La in Figure 2A) profiles exhibit a subtle deflection near 2.5 mm reflective of uptake via double medium diffusion (*sensu* [80]). Near 1 mm, numerous elements, especially Y, HREE, Sc, and U, alternatively exhibit a roughly 80% increase in concentrations over values at the cortical margin (e.g., Figure 2A–C), perhaps reflective of late diagenetic near-surface leaching (see Discussion).

Fe, Ba, Sr, and Mn each exhibit much flatter profiles at higher concentrations than all other elements (Figure 2C,D), with Fe exhibiting both the highest values and greatest range in variation of concentrations across the transect among these four elements. Unlike all other elements we investigated, Sc and U each exhibit “W-shaped” profiles with increasing cortical depth: after slowly decreasing from the cortical margin, each profile includes a broad, moderate peak in concentrations in the central portion of the middle cortex followed by steadily-increasing concentrations across the internal cortex (Figure 2B). Y exhibits the same profile shape as HREEs in the tibia of MOR 2598 (Figure 2C), indicating similar uptake behavior for these elements in this fossil.

### 4.3. NASC-Normalized REE Patterns

Spider diagrams of NASC-normalized REE concentrations reveal overall HREE enrichment in the bone as a whole (Figure 3B,C), but significant relative enrichment of LREEs within the external 250 μm of the cortex (Figure 4A). A ternary plot of Nd_N_-Gd_N_-Yb_N_ confirms this trend of relative HREE enrichment by revealing that a data point for the specimen as a whole plots more closely to the Yb corner (Figure 3C). Whereas there is no apparent Ce anomaly in the bone as a whole (Figure 3B), the external-most 250 μm of the cortex exhibits a modest positive Ce anomaly (seen as an upward deflection of the pattern at this element; Figure 4A). REE concentrations range from ~25 to 50 times NASC values in the external 250 µm of the cortex.

Substantial spatial heterogeneity in REE composition is evident in both a ternary plot of La_N_-Gd_N_-Yb_N_ (Figure 3D) and a spider diagram of individual laser runs compiled into the full transect (Figure 4B). Both of these figures reveal significantly greater LREE content in the external-most laser run compared to all other laser runs (i.e., variation exceeds two standard deviations), signifying variations in composition are largely controlled by cortical depth. In general, the bone becomes increasingly enriched in MREEs and HREEs relative to LREEs with increasing cortical depth, with transects through the middle and internal cortices exhibiting both similar magnitudes of REE enrichment and drastic relative enrichment in HREEs (Figure 4B). Proportionally, transects through these internal regions of the bone exhibit one to two orders of magnitude of enrichment in HREEs over LREEs, compared to just half an order of magnitude of HREE enrichment in the external-most transect.

All laser runs through the middle and internal cortices exhibit isolated peaks at gadolinium (Gd; Figure 4B), likely attributable to isobaric interference effects between LREE oxides and other ions likely present within the fossil (e.g., spectral overlap between Gd^157^ and BaF [81,82]). Most spider diagrams, especially those which separately plot data from the external cortex (e.g., Figure 4), also exhibit subtle peaks at europium (Eu) and holmium (Ho). These peaks impart a weak ‘M’ shape to the shale-normalized patterns, which most authors (e.g., [83] and references therein) attribute to influences of tetrad effects during uptake (also see Appendix A for further discussion on potential tetrad effects in MOR 2598).

### 4.4. (La/Yb)_N_ vs. (La/Sm)_N_ Ratio Patterns

The tibia of MOR 2598 exhibits a whole-bone average (La/Sm)_N_ value of 0.99 and a (La/Yb)_N_ of 0.26. These values signify modest HREE enrichment relative to many environmental water samples, dissolved loads, and sedimentary particulates. Specifically, these values place the bone within the compositional range of river waters, brackish estuary waters, and marine pore fluids (Figure 5A).

When plotted by individual laser runs (Figure 5B), REE ratios from the middle and internal cortices plot with similar (La/Sm)_N_ values to those seen in the external cortex but with consistently lower (La/Yb)_N_ values. This difference encompasses roughly one order of magnitude, on average. The most internal laser run exhibits the lowest (La/Yb)_N_ ratio (0.006), and all but two laser runs across the middle and internal cortices exhibit (La/Yb)_N_ ratios < 0.1. (La/Sm)_N_ ratios range between 0.8 and 1.5 and exhibit no apparent relationship with cortical depth.

### 4.5. REE Anomalies

Due to the concentrations of many trace elements in the middle cortex being so low that they fall below the lower detection limit (Appendix A), every anomaly examined exhibits major gaps in coverage through this region of the bone (Appendix A). Occasional instances of significantly higher neodymium (Nd) than praseodymium (Pr) concentrations also create gaps in the anomaly profiles. Whereas (Ce/Ce*)_N_ and La-corrected (Ce/Ce**)_N_ anomalies are absent at the outer cortex edge, (La/La*)_N_ anomalies are slightly negative in the external-most ~280 µm (Appendix A). All three of these anomalies exhibit substantial positive and negative fluctuations across the transect.

Although (Ce/Ce*)_N_ anomalies fluctuate from ~0.2 to 20 across the transect, they are largely positive throughout most of the internal half of the transect (Appendix A). This trend is not reflected, however, in the whole-bone (Ce/Ce*)_N_ average for the tibia, which is essentially absent (1.04; Table 1). (Ce/Ce*)_N_ values were also plotted against (Pr/Pr*)_N_ values (following [84]) to aid us in differentiating true, redox-related cerium anomalies from apparent anomalies induced by variations in (La/La*)_N_ anomalies. The majority of (Ce/Ce*)_N_ values from the external 1 mm of the bone plot near the lower margins of fields 3a and 4a (Figure 6), reflective of slightly negative La anomalies in the external cortex (in agreement with Appendix A). In contrast, anomaly values from inner regions of the cortex plot over a broad range encompassing every field of the diagram (Figure 6), indicative of substantial heterogeneity at the sub-millimeter scale in the middle and internal cortices. Within this broad spectrum, there are relatively few data points in fields 1 and 2b (Figure 6); regions of the internal cortex represented by these data points lack a Ce anomaly.

(La/La*)_N_ anomalies and La-corrected (Ce/Ce**)_N_ anomalies were also directly calculated (see Methods) to quantitatively assess these qualitative inferences. Unfortunately, as mentioned above, frequent drops in concentrations of LREEs below the detection limit severely limit coverage in these profiles. However, based on Appendix A, it is clear that (La/La*)_N_ anomalies are exclusively negative in the internal 29 mm of the cortex. The average (La/La*)_N_ value across this region is 0.20. (Ce/Ce**)_N_ anomalies are also almost exclusively negative in the middle and internal cortices, exhibiting a similarly low average (0.65) across this same span. The values of both of these anomalies fluctuate by roughly two orders of magnitude across the transect. As a whole, the tibia of MOR 2598 exhibits slightly positive (Ce/Ce**)_N_ and (La/La*)_N_ anomalies (1.10 and 1.12, respectively; Table 1), but these are each clearly biased by overweighting of data from the external cortex (caused by abundant missing data from internal regions of the bone, as discussed above). Plotting (Ce/Ce**)_N_ anomalies against U concentrations for each laser run yielded a poor correlation between these two redox-sensitive signatures (r^2^ = 0.29; Appendix A).

The yttrium/holmium (Y/Ho) ratios are slightly above chondritic (26; [85]) in the outer ~10 mm of the bone and the innermost ~7 mm of the transect, wherein they range ~20–300. Though data are sporadic through the middle cortex due to very low concentrations, ratios from this region form a broad swale below these peaks in the external and internal cortices (Appendix A). Specifically, the average of the Y/Ho ratios through the central 15 mm of the cortex (10–25 mm along the transect) is 35, and values across this region mostly fall between ~10 and 80. These spatial contrasts partially negate one another when data are averaged for the entire transect, which yields a slightly positive whole-bone average anomaly of 43 (Table 1).

## 5. Discussion

### 5.1. MOR 2598’s Paleoenvironmental and Taphonomic Context

Though limited taphonomic and stratigraphic data are available, the excellent preservation quality of MOR 2598 suggests it was protected by rapid burial postmortem. The full articulation of the hind limb, negligible signs of (ancient) weathering and abrasion, and excellent histological preservation (Figure 10B of [74]) each support the interpretation that burial took place within a few years postmortem, perhaps even just weeks after death [86]. However, the absence of the remainder of the skeleton implies this hind limb became disarticulated from the remainder of the carcass during brief subaerial decay, as well as the probable short-distance transport of the limb (from an upstream site of death) prior to burial (cf. [87,88]). The recovery of MOR 2598 from a channel sandstone [30] strongly suggests that: (1) decay primarily occurred subaqueously under oxygenated conditions (cf. [89]); (2) fluvial currents likely caused the separation of the limb from the body, and; (3) the carcass likely reached the “bloat” or active decay phase of postmortem decomposition for this to occur (cf. [90,91]). Ultimately, a lull in flow competency induced deposition and burial of the limb within the channel. Based on the great thickness of the succession of channel sandstone horizons from which MOR 2598 was recovered (7 m), it appears burial occurred within a well-established lowland channel rather than a recently-formed avulsion channel. This conclusion is consistent with prior interpretations of the Judith River Formation as generally representing lowland fluvial environments close to the coastline of the Western Interior Cretaceous Seaway (WIKS; [70,71]).

To briefly summarize, the available data reveal that this *Brachylophosaurus canadensis* individual (MOR 2598) died within or near a fluvial channel on the coastal lowlands. Its carcass experienced fairly brief decay within the channel, where currents eventually led to disarticulation of the left hind limb which was carried shortly downstream. Either an obstruction in the channel, a temporary lull in flow competency, or slowing of currents due to gradual channel broadening caused the deposition of the limb on the channel floor where it became quickly buried and fossilized within cross-stratified sands. Our trace element data provide illuminating insights into the ensuing diagenetic history of MOR 2598, which we now characterize in an effort to constrain geochemical pathways to cellular, soft tissue, and biomolecular preservation.

### 5.2. Reconstructing the Geochemical History of MOR 2598

The tibia of MOR 2598 exhibits low REE concentrations near the cortical margin (e.g., ~800 ppm for La; Appendix A) and a low whole-bone ∑REE (256 ppm) compared to many other bones from the Cretaceous period (Table 2), which have been found to possess ∑REE ranging from 1110 to 25,000 ppm [58,63,92,93,94,95]. Notably, however, these values each fall within the range of other dinosaur bones we have examined from the Cretaceous Hell Creek Formation [63,64] which have also been found to yield endogenous proteins [28,29,47]. Compared to those specimens from the Hell Creek Formation, MOR 2598 exhibits lower average concentrations of Fe (0.94 wt. %), Mn (834 ppm), and Y (190 ppm), a higher concentration of U (51 ppm), and similar concentrations of Sr (2499 ppm), Ba (1448 ppm), and Lu (3 ppm). Although these comparisons do not take into account differences in taxon, cortical width, histology, or diagenetic regimes, they still reveal that the tibia of MOR 2598 is (for most elements examined) less chemically altered than the majority of fossil bones of similar age. We have previously attributed such cases of minimal alteration to various sequestration processes limiting the availability of trace element ions in early-diagenetic pore fluids (e.g., complexation with humic acids and/or dissolved carbonates [96,97,98,99] and coprecipitation with phosphates in entombing sediments [100,101,102,103]), and those processes may also account for the modest alteration of MOR 2598 (see Appendix A for further discussion).

Ba, Fe, and Mn each exhibit flat concentration profiles (Figure 2C,D) probably indicative of incorporation into homogenously distributed, minute, secondary mineral phases, presumably barite, goethite, and Mn oxides [104]. Sr likely exhibits a similarly flat profile shape due to spatially homogenous substitution for Ca in bone hydroxyapatite [105,106]. In contrast to these more abundant elements, all REEs exhibit steep declines in concentrations from the cortical margin, with LREEs exhibiting the steepest declines and HREEs the shallowest (due to crystal–chemical controls based on ionic radius [107]). Meanwhile, MREE concentrations commonly drop below detection limit in the middle and internal cortices (Appendix A). These trends are typical of fossil bones which experienced relatively brief uptake largely by simple ‘external-to-internal’ diffusion and did not equilibrate with external pore fluids during diagenesis (cf. [60,63,64,68,108]). However, clear kinks in concentration profiles for many REEs and substantial, locally-restricted variations in their concentrations in the external cortex (e.g., La in Figure 2A) also indicate at least partial uptake via double medium diffusion (*sensu* [80]) through Haversian canals.

Spider diagrams reveal even proportions of REEs within the external-most cortex (Figure 4A) yet significant relative HREE enrichment throughout the bone as a whole (Figure 3B,C and Figure 4B). Relative HREE enrichment is especially evident in the internal cortex, where, for example, concentrations of Yb rise to more than double those of La (Figure 2A). These signatures are very similar to, but less pronounced than, those observed in a *Tyrannosaurus rex* femur recovered from an estuarine channel sandstone in the Hell Creek Formation [64]. As with that specimen, it is likely that such HREE enrichment reflects protracted trace element uptake from relatively HREE-enriched brackish waters and/or diagenetic pore fluids under oxidizing conditions. This interpretation is supported by the whole-bone composition of this specimen being similar to those of lowland river waters, estuarine waters, and marine pore fluids (Figure 5A) which typically exhibit such relative HREE enrichment [55,92,109]. Interestingly, these findings strongly suggest that the channel in which MOR 2598 was interred was likely tidally influenced, which in turn suggests that it was recovered from an estuarine channel, not a (strictly speaking) fluvial channel—an insight not apparent from the sedimentology/stratigraphy of the quarry.

Regarding redox regimes through diagenesis, at the whole-bone level, the tibia exhibits a slightly positive (Ce/Ce**)_N_ anomaly (1.10) reflective of a weakly oxidizing overall diagenetic history; this is consistent with the inferred burial setting having been an estuarine channel (see above). Generally high U concentrations throughout much of the cortex (Figure 2B and Table 1) and a relatively high average Sc concentration (59 ppm) corroborate this signal, as U and Sc enrichment have each been linked with uptake under oxidizing conditions [110,111,112]. The plotting of numerous data points from the middle and internal cortices in fields 3b and 4a of the (Ce/Ce*)_N_ vs. (Pr/Pr*)_N_ plot (Figure 6) also supports the presence of oxidizing conditions within the bone. However, both (Ce/Ce*)_N_ and (Ce/Ce**)_N_ anomalies are essentially absent at the cortical margin, whereas their values fluctuate considerably (both positively and negatively) throughout the middle and internal cortices (Appendix A). These contrasts, as well as the broad distribution of data points in Figure 6, demonstrate the presence of considerable spatial heterogeneity in redox conditions throughout the bone through diagenesis, especially in the middle and internal cortices.

Although the trace element anomaly profiles in Appendix A may seem somewhat stochastic, (Ce/Ce*)_N_ and (Ce/Ce**)_N_ values are generally positive and negative, respectively, across the inner half of the transect. If these trends are taken as reliable records of redox regimes during early diagenesis/fossilization, as all indications appear to support (see below), then these anomaly trends would signify that uptake occurred under prevailingly reducing conditions within the interior of the bone. The development of reducing microenvironments within fossil bones is relatively common [104] due to the release of iron and hydrogen sulfide from decaying organics within a dysaerobic, enclosed space [113]. However, high Sc concentrations in the internal cortex (Figure 2B) appear incompatible with this interpretation (as this should be a product of uptake under oxidizing conditions, as discussed above). We attribute this apparent “contrast” to temporal changes in redox conditions in this region of the bone through early diagenesis. Specifically, these conflicting signals could arise via the significant uptake of Sc in the internal cortex under initially oxidizing conditions followed by a shift to reducing conditions, during which latter time the internal cortex secondarily acquired positive (Ce/Ce*)_N_ anomalies (and Sc ions remained sequestered by, as in adsorped to, bone crystallites; also see the Appendix A for further discussion of the peculiar shapes of Sc and U concentration profiles in MOR 2598).

The redox scenario just described would necessitate a supply of significant amounts of Sc to the interior of the bone. This would have to be supplied by a pore fluid percolating through the medullary cavity (after the decay of blood and other internal organics), which would presumably also supply numerous other trace elements to the internal cortex. The concentration profiles of HREEs (e.g., Yb in Figure 2A), U (Figure 2B), Y, Lu, and the latter half of the MREE series (Appendix A) each exhibit increases toward the internal end of the transect (in the internal cortex), providing concrete evidence of uptake from a second diffusion front in the interior of the bone. That LREEs exhibit negligible rises in concentrations toward the internal end of the transect (e.g., La in Figure 2A) indicates that the majority of elements supplied by the pore fluid passing through the medullary cavity were mostly those with comparatively-modest to low diffusivities (based on [108]). This bias signifies that the pore fluid must have been a chemically ‘evolved’, highly-fractionated fluid which, based on the magnitude of select elemental enrichments in the internal cortex (e.g., Figure 2B), either flowed through the medullary cavity for an extended period of time or, more likely due to burial and compaction, became pooled there, allowing protracted uptake. It is also apparent that this pore fluid was likely not simply an HREE-enriched solution passing through during some later phase of late diagenesis because there are no clear signs of similar HREE enrichment in the external-most cortex (e.g., Figure 2A,B). Instead, the majority of elements exhibiting enrichment toward the internal end of the transect (e.g., U, Y, HREE) exhibit a subtle ‘plateau’ of stable concentrations in the outermost ~1 mm of the cortex followed by a ~80% increase near ~1.5–2 mm (Figure 2A–C). We interpret this pattern to reflect modest leaching of trace elements from the outermost ~1 mm of the external cortex, most likely during late diagenesis and under slightly oxidizing conditions (based on the weakly positive (Ce/Ce*)_N_ anomalies in this region; Appendix A). This conclusion may also be supported by: (1) the common presence of negative (La/La*)_N_ anomalies in the outermost ~500 µm of the external cortex (potentially reflective of near-surface loss of La; Appendix A), and; (2) a lack of a correlation between U concentrations and (Ce/Ce**)_N_ anomalies for each laser run (r^2^ = 0.29; Appendix A), implying uptake of U and REEs over differing timescales [112].

The absence of more major signs of leaching or late-diagenetic trace element uptake at the cortical edge (Figure 2), as well as the retention of clear evidence of spatiotemporal changes in pore fluid compositions (described in the last few paragraphs), imply that late-diagenetic overprinting of trace element signatures in MOR 2598 was not substantial, and, therefore, that the tibia at least partially retains early-diagenetic signatures. Indeed, there are numerous signs of relatively brief interaction with pore fluids. For example, a spider diagram (Figure 4B) and ternary plot (Figure 3D) of REE proportions by individual laser runs each reveal clear signs of significant fractionation during uptake from circum-neutral pH surface/groundwaters (cf. [114]) in the form of increasing relative LREE depletion/HREE enrichment with increasing cortical depth (as in, e.g., [59,107,115]). These fractionation effects are also evident from significant spatial variations in (La/Yb)_N_ ratios (Figure 5B), a positive (La/La*)_N_ anomaly and Y/Ho ratio for the bone as a whole (1.12 and 43, respectively; cf. [60]), and differing concentration profile shapes (contrast Figure 2A,B) for U and REE (which should otherwise have similar shapes due to their similar diffusivities [108]). Similarly-high Y/Ho ratios in the external and internal cortices (Appendix A) further imply fractionation also occurred during uptake from the chemically ‘evolved’ pore fluid pooled in the medullary cavity.

To review, our trace element data thus identify that after brief subaqeous decay and transport down a fluvial stream, the hind limb of MOR 2598 became fossilized after burial in a sandy, oxic, estuarine channel (Figure 7). Its bones experienced relatively brief primary uptake of REEs and other trace elements from circum-neutral pH, HREE-enriched, and potentially brackish channel waters and groundwaters under oxidizing conditions during early diagenesis. Comparatively slower percolation of pore fluids through the medullary cavity of the tibia than around its exterior led to the development of reducing conditions inside the bone. Recent erosion, typical of that in desert/badlands environments from which most fossil bones are recovered, re-exposed MOR 2598 to oxidizing conditions and caused minor leaching of trace elements from the outermost ~1 mm of the cortex but had no significant effects on the overall chemistry of the bone. As a result, the specimen remains modestly altered compared to others of similar age.

### 5.3. Insights into Molecular Taphonomy from Comparative Geochemistry

MOR 2598 is only the third vertebrate fossil of pre-Cenozoic age to have both yielded endogenous protein and have its geochemical history characterized through trace element analyses. Both other specimens in this short list, namely *Tyrannosaurus rex* MOR 1125 [28,29,64] and *Edmontosaurus annectens* SRHS-DU-231 [47,63], are also large nonavian dinosaurs recovered from Late Cretaceous strata in Montana, USA. Adding MOR 2598 into this comparative framework reveals both geochemical similarities to, and differences from, these two other specimens which: (1) bolster prior hypotheses about protein preservation pathways; and (2) add new insights into the complexity of post-burial diagenetic alterations which biomolecules can withstand.

Taken as a whole, the biostratinomic history of *Brachylophosaurus* MOR 2598 is quite similar to that which we recently explicated for *T. rex* MOR 1125 [64]. After death and a short period of (likely subaqueuous) decay, brief fluvial transport brought each specimen into coastal estuaries along the western coast of the WIKS where they were rapidly buried in sandy estuarine channels, and throughout this history, the bones of each specimen were acquiring trace elements from brackish, HREE-enriched surface and groundwaters (this study and [64]). As at the MOR 1125 quarry [64], early cementation of the sediments entombing MOR 2598 appears to have limited trace element uptake by the fossil bones, allowing them to exhibit minimal alteration at the elemental level. This is evident in the low ∑REE of the tibia compared to many other bones of Cretaceous age (as discussed above), as well as its steep declines in REE concentrations from the cortical margin (Figure 2A) and very low concentrations of elements with ionic radii similar to that of Ca^2+^ (i.e., MREE) in the middle cortex (Appendix A). Thus, MOR 2598 adds further support to the assertions of Schweitzer [116], Herwartz et al. [59], and Ullmann et al. [64] that: (1) early-diagenetic cementation of sediments can effectively thwart protracted decay and chemical alteration of bones after burial (presumably by minimizing exposure to percolating groundwaters and the exogenous microbes they carry with them), and; (2) this diagenetic pathway also facilitates rapid molecular stabilization (presumably via the iron free-radical-induced molecular crosslinking mechanism elucidated by Boatman et al. [53]). Fossils from the Standing Rock Hadrosaur Site (SRHS; [47,63,117]) demonstrate that rapid burial in fine-grained sediments with low-permeability and/or encasement in early-diagenetic concretion can similarly hinder the decay of endogenous cells, tissues, and their component biomolecules.

As for SRHS-DU-231 [63] and MOR 1125 [64], MOR 2598 exhibits high Sc enrichment and a slightly positive whole-bone (Ce/Ce**)_N_ anomaly (Table 1) reflective of a generally oxidizing diagenetic history. Although this pattern superficially supports the proposition by Wiemann et al. [54] that oxidizing depositional environments may be more favorable settings for molecular preservation than reducing environments (perhaps due to greater release of crosslink-catalyzing iron free radicals; cf. [53]), it is clear that both MOR 2598 and bones from SRHS also experienced reducing conditions during diagenesis. This is evident from consistently-positive (Ce/Ce*)_N_ anomalies in the internal cortex of the tibia of MOR 2598 (Appendix A) and external cortices of many SRHS bones (Figure 6 of [63]). Microbial decay of organics (including those within bones) is known to decrease local pH and create dysoxic to anoxic conditions, thereby eliciting the production of reducing conditions [118], especially within confined microenvironments such as those within and around a carcass in compacted or low-permeability sediments (e.g., [50,119]) or within the internal pore spaces of bones (e.g., [104,120,121]). It is also well known that reducing early diagenetic conditions do not preclude exquisite morphologic and biochemical preservation of structural soft tissues, perhaps due to induced dysoxia/anoxia (e.g., [122,123,124,125]). Thus, there are reasons that fossil bones preserved under largely-reducing conditions may still yield original molecules.

Though the acidic pH that would have temporarily accompanied reducing conditions within the medullary cavity of this specimen may seem (at face value) preclusive to molecular preservation, weak acidity has been implicated in the rapid nucleation of inert, protective, microcrystalline goethite crystals within ‘osteocytes’ and ‘blood vessels’ recovered from fossil bones [4,53]. For this reason, initial redox conditions in the immediate ~48 h after death may ultimately be the most critical, as it has been demonstrated that inter- and intra-molecular crosslinking (i.e., stabilization) reactions can operate in this brief timeframe, promoting equilibration with the early-diagenetic environment which may then persist through fossilization and late diagenesis [53]. However, we must note that actualistic studies examining molecular stability through temporal shifts in redox regimes would be necessary to further evaluate this hypothesis. Regardless, our recognition of varied pH and redox conditions over time within the tibia of MOR 2598 is thus not incompatible with these prior studies, but rather augments them by revealing that biomolecular remains may survive multiple changes in redox regimes through diagenesis (Figure 7).

In particular, as discussed above, the bones of MOR 2598 experienced two comparatively ‘extra’ diagenetic events which MOR 1125 and bones from SRHS did not [63,64]: (1) protracted trace element uptake from stagnant pore fluids ‘pooled’ in the central medullary cavity (evidenced by elevated concentrations of HREEs, U, Sc, Y, and Lu in the internal cortex; Figure 2, Appendix A), and; (2) modest late-diagenetic leaching of the cortical surface (evidenced by negative (La/La*)_N_ anomalies and reduced concentrations of many trace elements in the outermost ~0.5–1 mm; Figure 2 and Appendix A). Despite this complex diagenetic history, cortical bone from the femur of MOR 2598 has yielded numerous microstructures retaining endogenous peptide sequences of multiple proteins (namely collagen, actin, tubulin, histones, myosin, and tropomyosin [18,30,45]). This fact indicates that processes which stabilized diagenetiforms within this specimen in the initial hours to days postmortem imparted remarkable long-term resiliency. Novel experiments by Schwietzer et al. [4] and Boatman et al. [53] have begun to shed light on how this may occur, but testing of more fossils and further actualistic studies are needed to fully resolve the endurance of diagenetiforms under the wide array of physicochemical/thermodynamic regimes of natural diagenetic environments.

Finally, all three specimens examined in this discussion (MOR 2598, SRHS-DU-231, and MOR 1125) also each exhibit abundant signals of fractionation of REEs during uptake, which when combined with their low ∑REE content indicate at least partial retention of early-diagenetic trace element signatures. These signs include positive whole-bone Y/Ho anomalies (Table 1, and Table 1 in [63,64]), negative (La/La*)_N_ anomalies in the middle and internal cortices (Figures S1 and S2 of [64]), steeper concentration profiles for LREEs than HREEs (Figure 2A, Figure 2 of [63], and Figure 4A of [64]), and increasing relative-HREE enrichment with cortical depth (Figure 4B, Figure 4 of [63], and Figure 6B of [64]). As suggested by Ullmann et al. [64], the retention of such early-diagenetic signatures may constitute a useful proxy for molecular recovery potential because it indicates a specimen has avoided protracted interactions with any late-diagenetic pore fluids (e.g., phreatic groundwaters) which could plausibly cause hydrolysis and other decay processes.

## 6. Conclusions

Synthesizing our results with those of other recent experimental and actualistic studies in molecular taphonomy leads us to conclude the following:By allowing the quick characterization of spatial patterns of diagenetic alteration within a fossil, trace element analyses constitute a useful and effective means of screening fossil tissues prior to paleomolecular analyses;Retention of early-diagenetic trace element signatures may constitute a useful proxy for molecular recovery potential because it indicates a specimen has avoided protracted interactions with any late-diagenetic pore fluids;Although burial in coarse, permeable sediments in oxic environments appears conducive to molecular preservation, it is also possible for fossils preserved in reducing paleoenvironments to yield endogenous molecules. This dichotomy suggests that the presence of oxidizing conditions in the initial ~48 h postmortem may be more key to molecular preservation than the redox state of the final setting of burial;Rapid burial in fine-grained sediments with low permeability, encasement in early-diagenetic concretion, and/or early-diagenetic cementation of entombing sediments can effectively thwart protracted decay and chemical alteration of bones (and their component cells, tissues, and biomolecules) by minimizing exposure to percolating groundwaters and the exogenous microbes they carry with them, and;Biomolecular remains may survive multiple changes in redox regimes through diagenesis, and this indicates that processes which stabilize biomolecules in the initial hours to days postmortem can impart remarkable long-term resiliency.

While these insights are obviously enlightening, it must be reiterated that they primarily derive from fossils from just three Cretaceous localities. In agreement with Schroeter et al. [126], we propose that paleomolecular and trace element analyses on Paleogene and Miocene fossils are direly needed to close the long window of the Cenozoic for which protein preservation has yet to be explored. Based on all data currently available, it seems very likely that future studies will continue to broaden the suite of depositional and diagenetic circumstances known to be conducive to molecular preservation.

## Figures and Tables

**Figure 1 biology-11-01177-f001:**
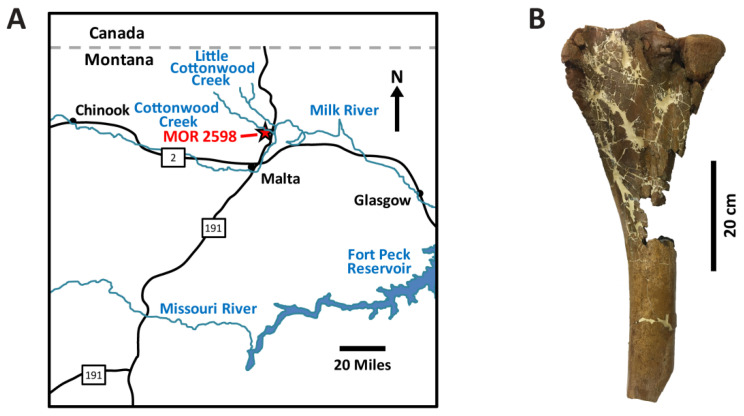
(**A**) Map showing the locality from which MOR 2598 was recovered in Phillips County, Montana. (**B**) Left tibia of MOR 2598 examined in this study, shown in lateral view. Map redrawn and modified from [75].

**Figure 2 biology-11-01177-f002:**
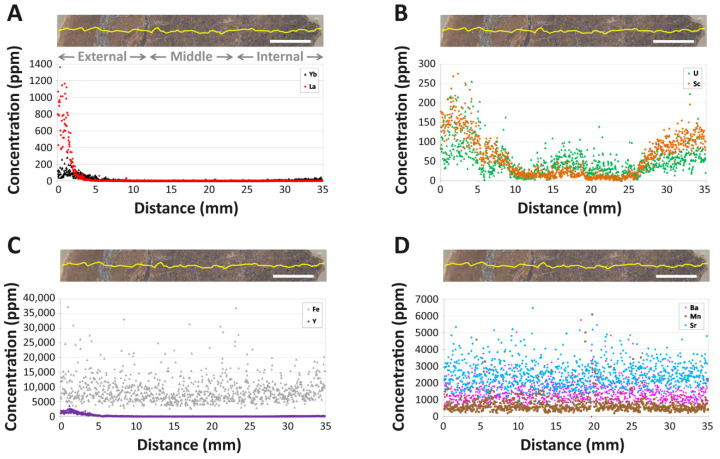
Intra-bone concentration gradients of various elements in the tibia of MOR 2598. (**A**) Lanthanum (La) and ytterbium (Yb). (**B**) Scandium (Sc) and uranium (U). (**C**) Iron (Fe) and yttrium (Y). (**D**) Barium (Ba), manganese (Mn), and strontium (Sr). Note the different concentration scales for each panel. The laser track is denoted by the yellow line in each bone cross section. Gray text labels in (**A**) span the approximate regions considered as the ‘external’, ‘middle’, and ‘internal’ cortices. Scale bars, in white over bone images, each equal 5 mm.

**Figure 3 biology-11-01177-f003:**
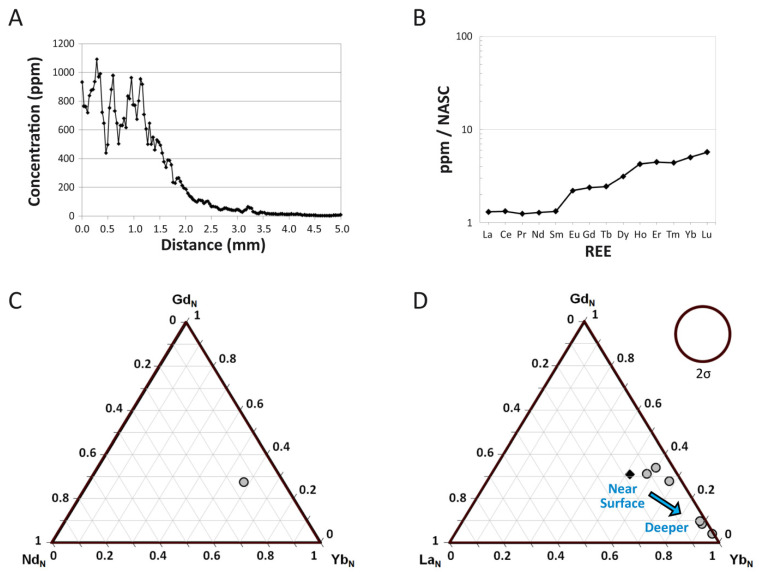
REE composition of the tibia of MOR 2598. (**A**) Three-point moving average profile of La concentrations in the outermost 5 mm of the bone. (**B**) Average NASC-normalized REE composition of the fossil specimen as a whole. (**C**,**D**) Ternary diagrams of NASC-normalized REE. (**C**) Average composition of the bone. (**D**) REE compositions divided into data from each individual laser transect (~5 mm of data each). Compositional data from the transect that included the outer bone edge is denoted by a dark diamond; all other internal transect data are indicated by gray circles. The 2σ circle represents two standard deviations based on ± 5% relative standard deviation.

**Figure 4 biology-11-01177-f004:**
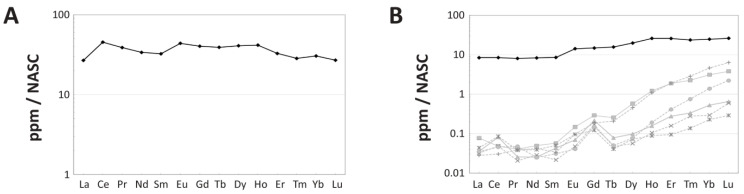
Spider diagrams of intra-bone NASC-normalized REE distribution patterns within the tibia of MOR 2598. (**A**) Average composition of the outermost 250 µm of the cortex, demonstrating substantially greater LREE and MREE content in the outermost cortex compared to in the bone as a whole (**B**). (**B**) Variation in compositional patterns by laser transects. The pattern which includes the external margin of the bone is shown in black, those from deepest within the bone by dotted, light-gray lines, and all other analyses in between by solid, dark-gray lines.

**Figure 5 biology-11-01177-f005:**
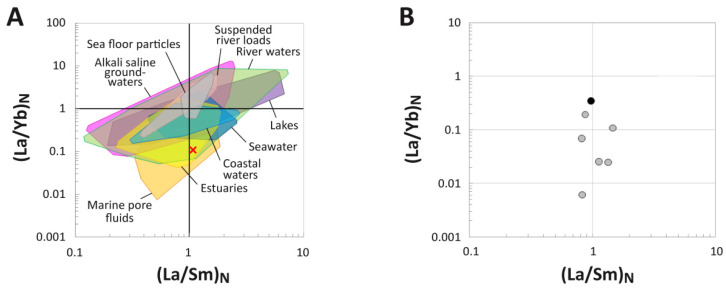
(La/Yb)_N_ and (La/Sm)_N_ ratios of the tibia of MOR 2598. (**A**) Comparison of the whole-bone average (La/Yb)_N_ and (La/Sm)_N_ ratios of the fossil to ratios from various environmental waters and sedimentary particulates. Literature sources for environmental samples are provided in the Appendix A. (**B**) REE compositions of individual laser transects expressed as NASC-normalized (La/Yb)_N_ and (La/Sm)_N_ ratios. The transect including the external bone margin is denoted by the black symbol, whereas all other (internal) transects are represented by gray symbols.

**Figure 6 biology-11-01177-f006:**
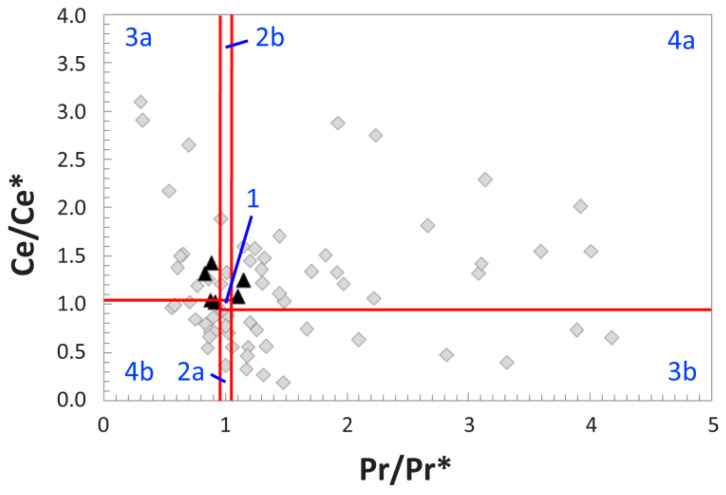
(Ce/Ce*)_N_ vs. (Pr/Pr*)_N_ plot (after [84]) of five-point averages along the transect across the cortex of MOR 2598 recorded via LA-ICPMS. Separate fields (labeled by blue text) are as follows: 1, neither Ce nor La anomaly; 2a, no Ce and positive La anomaly; 2b, no Ce and negative La anomaly; 3a, positive Ce and negative La anomaly; 3b, negative Ce and positive La anomaly; 4a, negative Ce and negative La anomaly; 4b, positive Ce and positive La anomaly. Measurements from the outer 1 mm of the external cortex are plotted as black triangles, and all measurements from deeper within the bone are plotted as gray diamonds. (Ce/Ce*)_N_ and (Pr/Pr*)_N_ anomalies calculated as in the Materials and Methods section of the text.

**Figure 7 biology-11-01177-f007:**
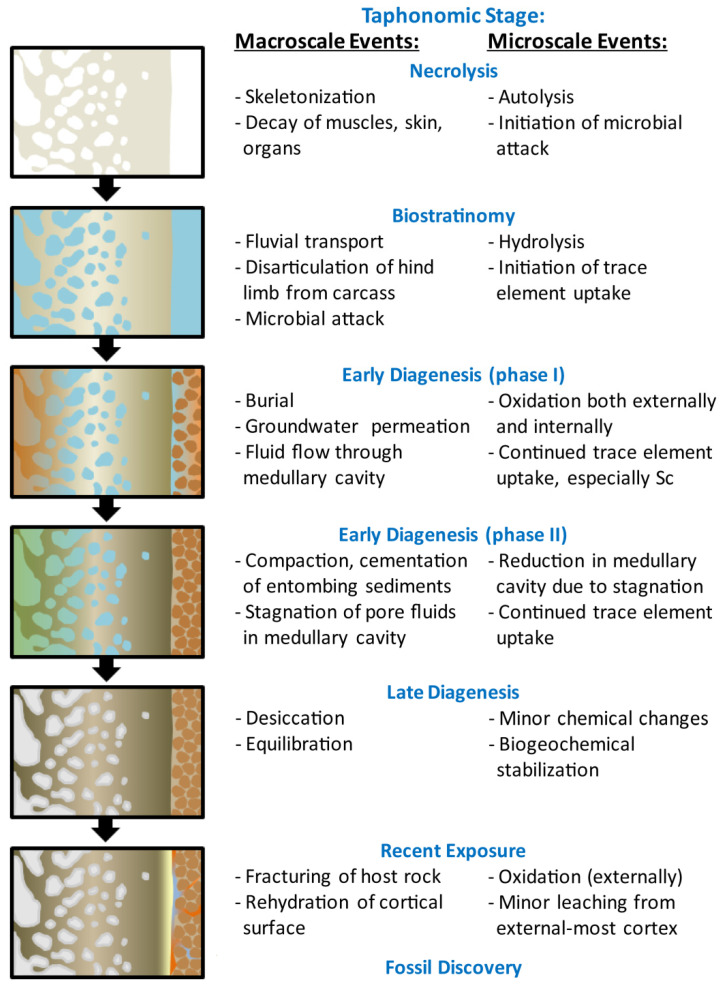
Reconstruction of the diagenetic history of *Brachylophosaurus canadensis* MOR 2598, summarizing macroscale and microscale events effecting its skeletal elements after death. Events are classified into approximate taphonomic stages, with a visual portrayal of the diagenetic processes affecting the bones during each stage shown at left.

**Table 1 biology-11-01177-t001:** Average whole-bone trace element composition of the left tibia of *Brachylophosaurus canadensis* MOR 2598. Numbers presented are averages of all transect data acquired across the cortex. Iron (Fe) is presented in weight percent (wt. %); all other elements are in parts per million (ppm). Absence of (Ce/Ce*)_N_, (Pr/Pr*)_N_, (Ce/Ce**)_N_, and (La/La*)_N_ anomalies occurs at 1.0, and these anomalies were calculated as in the Materials and Methods. The Y/Ho value reflects this mass ratio.

Element	Concentration
Sc	59.23
Mn	834
Fe	0.94
Sr	2499
Y	190
Ba	1448
La	40.65
Ce	88.61
Pr	9.86
Nd	35.26
Sm	7.41
Eu	2.61
Gd	12.41
Tb	2.09
Dy	17.33
Ho	4.44
Er	15.26
Tm	2.21
Yb	15.42
Lu	2.62
Th	0.18
U	51.13
∑REE	256
(Ce/Ce*)_N_	1.04
(Pr/Pr*)_N_	0.95
(Ce/Ce**)_N_	1.10
(La/La*)_N_	1.12
Y/Ho	42.85

**Table 2 biology-11-01177-t002:** Summary of the REE composition of the left tibia of *Brachylophosaurus canadensis* MOR 2598. Qualitative ∑REE content is based on the value shown in Table 1 (256 ppm) in comparison to values from other Mesozoic bones (as listed in the main text). Abbreviations: DMD, double medium diffusion *sensu* [80]; LREEs, light rare earth elements.

**Clear DMD Kink for LREE?**	**Relative Noise in Outer Cortex for La**	**REE Suggest Flow in Marrow Cavity?**	**Relative ∑REE Content (Whole Bone)**	**Relative U Content (Whole Bone)**	**Relative Porosity of the Cortex**
Yes	Moderate	Yes	Low	Moderate	Low

## Data Availability

All data generated by this study are available in this manuscript and the accompanying Appendix A.

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
