# Peer review of "Taphonomic and Diagenetic Pathways to Protein Preservation, Part II: The Case of Brachylophosaurus canadensis Specimen MOR 2598"

_biology, 2022, doi:10.3390/biology11081177_

Round 1

Reviewer 1 Report

I have no real problems with this paper. It was interesting to read, thought provoking and provides signposts to future valuable work. Excellent.

There are a couple of points I have highlighted (literally) in the annotated manuscript.

1) Lines 95-96: What exactly does this mean? This sounds like weathering out of the surrounding sediments but I don't think that is what you mean.

2) Lines 168-170: It is certainly worth stressing here that at the time of its incorporation in this sediment there was some soft tissue (skin? cartilage? ligaments? so therefore probably collagen) present and maintaining approximate anatomical positions.

3) Figure 2C: It is not easy to distinguish the different elements here.

4) Line 289: Perhaps worth explaining this a little further or adding a suitable reference.

5) Line 390: Is this supported by histological images? Reference [70] has an image of a cross-section of this bone MOR 2598 in Figure 10B, although this is not made clear in the figure caption. Please refer specifically to "Figure 10B in [70]" and any other places this histology may have been published.

6) Line 393: This is pure speculation on the part of the reviewer but a plausible scenario is that this carcass was washed downriver and deposited in an estuarine environment, or became mired in soft sediment and was preyed upon by predators/scavengers. In either case it seems likely that weeks is a more probable timeframe than years for becoming encased in sediments (especially if they were soft). I think the authors are way ahead of me on this but did not make it explicit.

7) Lines 484-485: Are there any clues about the timescales we are talking about here?

8) Line 503: My understanding of non-avian dinosaur anatomy is a bit shaky. Would the tibial medullary cavity be air filled or fat/cellular mass filled during life? Would this remain void post mortem? If the medullary cavity is organic-rich then perhaps anoxic conditions arising from chemical degradation/oxidation of lipids/proteins/mucopolysaccharides might be expected in an unbroken bone.

9) Line 554: See comment 8).

10) Line 587" Can the authors suggest a plausible mechanism for this stabilisation?

11) Line 590: Again, a more detailed mechanism would be useful.

12) Figure 7: Microbial attack on the soft tissues or the mineralised collagen?

13) Line 600: This statement is counterintuitive, so perhaps a little more reasoning at this point would be warranted.

14) Line 612: I find the idea that reducing conditions, and low pore-water mobility alone are sufficient to preserve biomolecules over deep time to be implausible. However, personal incredulity is not a sound basis for rejection of scientific findings. After all, proteins do survive within amber over deep time (feathers).

15) Line 626-627: In fact these changes may be a prerequisite for preservation.

One final comment. Reducing microniches in bones surrounded by, or saturated with, water are not so very unusual. I have seen pyrite framboids in very recent bones washed up on the beach. These have no visible impregnation with iron salts but the presence of pyrite demonstrates it is there. I suspect chemical decomposition of organic residues other than collagen is sufficient to depress the Eh and probably also the pH to begin mobilising Ca and P from poorly crystalline regions of the tissue.

Author Response

Reviewer 1, Comment 1: “I have no real problems with this paper. It was interesting to read, thought provoking and provides signposts to future valuable work. Excellent.”

Response: Thank you!

---

Reviewer 1, Comment 2: “Lines 95-96: What exactly does this mean? This sounds like weathering out of the surrounding sediments but I don't think that is what you mean.”

Response: Actually, the reviewer’s interpretation is pretty close to our intended meaning. As reviewed by Trueman (2007), trace elements in sedimentary pore fluids are sourced from physical and chemical weathering of rocks within surficial soils or exposed at the ground surface. Specifically, trace element ions can be solubilized from surrounding sediments via oxidation, reduction, and/or dissolution (Trueman 2007; Kowal-Linka and Jochum 2015). To clarify this, we have rephrased the beginning of the sentence marked by the reviewer to begin as follows: “After being solubilised from surrounding sediments by percolating groundwaters via oxidation, dissolution, and other processes, trace element ions...”

---

Reviewer 1, Comment 3:  “Lines 168-170: It is certainly worth stressing here that at the time of its incorporation in this sediment there was some soft tissue (skin? cartilage? ligaments? so therefore probably collagen) present and maintaining approximate anatomical positions.”

Response: While the presence of connective soft tissues (e.g., ligaments) ‘outside’ the bones at the time of burial does not seem particularly significant to the long-term preservation of soft tissues ‘inside’ them, we agree it is an accurate observation which we could include. Therefore, we have included this interpretation as a new sentence in the Taphonomic and Geologic Context section. It reads as follows: “Articulation of these skeletal elements implies they were still joined by connective tissues (i.e., ligaments) at the time of burial.”

---

Reviewer 1, Comment 4: “Figure 2C: It is not easy to distinguish the different elements here.”

Response: The two colors used for each dataset in this figure panel, purple and gray, are not close in color coding (i.e., RGB values), so they should be readily distinguishable from one another visually. Additionally, the points for Y were plotted as diamonds while those for Fe were plotted in a different shape, triangles. Perhaps the reviewer felt it was difficult to see the gray Fe data points? In short, we are not certain what was difficult for the reviewer to distinguish. The color and symbol scheme employed in this panel matches that which we used in Figure 4C of our “Part I” study of T. rex MOR 1125 published last year in this same Special Issue of Biology (Ullmann et al. 2021). Therefore, in the absence of a more specific recommendation from the reviewer, we feel it is best to keep the visual scheme of this figure panel matching that of Ullmann et al. (2021) so that this graph of Fe and Y in each paper can be easily compared with one another.

---

Reviewer 1, Comment 5: “Line 289: Perhaps worth explaining this a little further or adding a suitable reference.”

Response: In our manuscript, we cite the referral of isolated Gd peaks to isobaric interference effects to Kemp and Trueman (2003) and Smirnova et al. (2006). We believe these two papers are sufficient for citation purposes, but agree with the reviewer that the concept of isobaric interference can be explained in greater detail for readers less familiar with ICPMS. To help make this concept more accessible to all readers, we have revised the end of the sentence to provide more detail and an example (from Smirnova et al. 2006). It now ends as follows: “...likely attributable to isobaric interference effects between LREE oxides and other ions likely present within the fossil (e.g., spectral overlap between Gd157 and BaF [77,78]).”

---

Reviewer 1, Comment 6: “Line 390: Is this supported by histological images? Reference [70] has an image of a cross-section of this bone MOR 2598 in Figure 10B, although this is not made clear in the figure caption. Please refer specifically to "Figure 10B in [70]" and any other places this histology may have been published.”

Response: Accepted. We have revised the citation in line 390 to Schweitzer et al. (2016) to specify it to their figure 10B. This is the only citation to that fact from that reference, so the change was only needed in this single line.

---

Reviewer 1, Comment 7: “Line 393: This is pure speculation on the part of the reviewer but a plausible scenario is that this carcass was washed downriver and deposited in an estuarine environment, or became mired in soft sediment and was preyed upon by predators/scavengers. In either case it seems likely that weeks is a more probable timeframe than years for becoming encased in sediments (especially if they were soft). I think the authors are way ahead of me on this but did not make it explicit.”

Response: We agree with the reviewer’s speculation about the probable sequence which MOR 2598 experienced to become buried. As it is not possible at this time to constrain the timing nor precise sequence of events further, we believe the sentence marked by the reviewer already nicely summarizes what we can say with confidence from the available taphonomic evidence, so we have not made any changes based on this comment.

---

Reviewer 1, Comment 8: “Lines 484-485: Are there any clues about the timescales we are talking about here?”

Response: A short answer to this question is “no”, because trace element signatures can provide clues as to the relative sequence of events, such as in terms of ‘early’ and ‘later’ diagenesis, but generally not about the absolute timing or lengths of diagenetic events (such as interactions with pore fluids). Variations in pore fluid flow rate, elemental concentration gradients over time, and in uptake behavior related to histologic structure (see, e.g., Hinz and Kohn 2010; Kohn and Moses 2013) ultimately cause uptake to occur at varied rates, and this generally precludes any attempts to temporally constrain the timescales of diagenetic events in precise numbers. Thus, we instead use the more generic terms ‘early’ and ‘late’ diagenesis throughout the manuscript.

---

Reviewer 1, Comment 9: “Line 503: My understanding of non-avian dinosaur anatomy is a bit shaky. Would the tibial medullary cavity be air filled or fat/cellular mass filled during life? Would this remain void post mortem? If the medullary cavity is organic-rich then perhaps anoxic conditions arising from chemical degradation/oxidation of lipids/proteins/mucopolysaccharides might be expected in an unbroken bone.”

Response: The reviewer raises a great point here that we took knowledge of bone anatomy by the reader for granted in this sentence. To help readers less familiar with osteology understand our intended meaning, we have added a parenthetical phrase to the sentence clarifying that the diagenetic pore fluid would occupy interior space formally occupied by vascular tissue, blood, and other organics during life. This sentence now reads as follows: “This would have to be supplied by a pore fluid percolating through the medullary cavity (after the decay of blood and other internal organics), which would presumably also supply numerous other trace elements to the internal cortex.”

---

Reviewer 1, Comment 10: “Line 554: See comment 8).”

Response: Please see our response to Reviewer 1’s Comment 8 above, which we think satisfies the reviewer’s inquiry in this line as well.

---

Reviewer 1, Comment 11: “Line 587" Can the authors suggest a plausible mechanism for this stabilisation?”

Response: Though the precise mechanism of stabilization at the molecular level which occurred within the bones of MOR 2598 remains to be clarified, we hypothesize it was likely the mechanism elucidated by the experiments of Boatman et al. (2019), which we mentioned briefly in our Introduction section. Those authors found that oxidation of decaying soft tissues and cells releases iron free radicals which can induce inter- and intramolecular crosslinking, thus creating a natural pathway to tissue fixation and long-term biomlecular stability. We have added this hypothesized mechanism as a parenthetical addition to the end of the sentence in line 587. It now ends as follows: “...this diagenetic pathway also facilitates rapid molecular stabilization (presumably via the iron free radical-induced inter- and intramolecular crosslinking mechanism elucidated by Boatman et al. [53]).”

---

Reviewer 1, Comment 12: “Line 590: Again, a more detailed mechanism would be useful.”

Response: This comment is inherently linked with the previous comment by the reviewer, as both refer to (likely) the same mechanism. Given this overlap, we thought it best to add details about the potential mechanism (that elucidated by Boatman et al. 2014) in only one sentence in this paragraph of the text, rather than creating redundancy by adding it in both lines marked by the reviewer. Please see our response to the previous comment to see how we added the mechanism into line 587.

---

Reviewer 1, Comment 13: “Figure 7: Microbial attack on the soft tissues or the mineralised collagen?”

Response: This bullet in the figure is referring to microbial attack on either, or both, components, though most workers use this term in reference to microbial exploitation of the organic soft-tissue components. Since the short phrase “microbial attack” is a general term which can encompass attack on both components, we have elected to leave the wording of this bullet in the figure as it is and did not change it.

---

Reviewer 1, Comment 14: “Line 600: This statement is counterintuitive, so perhaps a little more reasoning at this point would be warranted.”

Response: We agree that the hypothesis of Wiemann et al. (2018) that oxidizing conditions can actually promote molecular preservation is counterintuitive, and those authors also acknowledged the counter-intuitiveness in their paper. Based on the current understanding of the field, it appears that oxidizing conditions can stimulate the release of iron free radicals which, as explained in our response to Reviewer 1’s Comment 11 above, can incite molecular stabilization via crosslinking (Boatman et al. 2019 = our Ref. #53). We have added a parenthetical phrase to the sentence in line 600 summarizing this linkage, which reads as follows: “...that oxidizing depositional environments may be more favourable settings for molecular preservation than reducing environments (perhaps due to greater release of crosslink-catalyzing iron free radicals; cf. [53]), it is clear...”

---

Reviewer 1, Comment 15: “Line 612: I find the idea that reducing conditions, and low pore-water mobility alone are sufficient to preserve biomolecules over deep time to be implausible. However, personal incredulity is not a sound basis for rejection of scientific findings. After all, proteins do survive within amber over deep time (feathers).”

Response: The reviewer is correct that select fossil specimens, including bones, which were preserved largely under reducing conditions during fossilization have been identified to yield ancient molecular signatures (e.g., Ullmann et al. 2020b; Voegele et al. in this same Special Issue of Biology). More importantly, we agree that these conditions alone constitute an incomplete explanation of the preservation pathway (i.e., they do not explicitly identify the chemical reactions which took place leading to biomolecular stabilization), and our sentence in line 612 acknowledges this limitation. We do not say that these conditions “alone are sufficient to preserve biomolecules” (in the reviewer’s words), but rather that specimens preserved in these circumstances “may still yield original molecules.” Thus, we believe no change is needed to this sentence, and thus have made no changes to the text based on this comment.

---

Reviewer 1, Comment 16: “Line 626-627: In fact these changes may be a prerequisite for preservation.”

Response: The reviewer advances an intriguing idea here, which is difficult to evaluate given the current data available from this study and from other similar studies by the molecular paleontology community. This is definitely a subject in need of further exploration in future studies. Since no single study can likely answer this question yet, we have made no changes to the text based on this comment, and instead look forward to further evaluating this idea in the future.

---

Reviewer 1, Comment 17: “One final comment. Reducing microniches in bones surrounded by, or saturated with, water are not so very unusual. I have seen pyrite framboids in very recent bones washed up on the beach. These have no visible impregnation with iron salts but the presence of pyrite demonstrates it is there. I suspect chemical decomposition of organic residues other than collagen is sufficient to depress the Eh and probably also the pH to begin mobilising Ca and P from poorly crystalline regions of the tissue.”

Response: We agree with the reviewer on all accounts, and agree that these are relatively ‘well-established’ decay and preservation processes known to operate in natural environments. Production of reducing conditions within bones is indeed known to occur, and given the right conditions it is both likely and common (e.g., Pfretzschner 2001; Wings 2004). To acknowledge that development of a reducing microenvironment within the medullary cavity of the tibia of MOR 2598 is a fairly ‘typical’ diagenetic occurrence, we have added a new sentence to the second paragraph on redox conditions in the Discussion. This new sentence, inserted at line 491, reads as follows: “Development of reducing microenvironments within fossil bones is relatively common (Wings 2004) due to release of iron and hydrogen sulfide from decaying organics within a dysaerobic, enclosed space (Brett and Baird 1986).”

---

Reviewer 1, Comment 18: Line 617-621: “This for me is the key part. But I am surprised that phosphatisation has not been suggested. After all, phosphates are potentially abundant in a buried bone undergoing early diagenesis and phosphatisation is known to promote preservation of physical structures in exquisite detail. e.g. McNamara et al. 2009 Organic preservation of fossil musculature with ultracellular detail; and Gueriau et al. 2020 Oxidative conditions can lead to exceptional preservation through phosphatization. Similarly, other protein preservation through deep time is found in bones, bivalves and eggshells where protein fragments are thought to be encased and surrounded in protective mineral.”

Response: First, thank you for referring us to Gueriau et al. (2020) – we were not aware of this publication, and it definitely presents a great advancement forward for relevant analytical methodologies. We agree that phosphatization has been a commonly-advanced pathway to the preservation of ultrastructural tissues, such as muscle fibers in McNamara et al. (2009) and Kellner (1996), and precipitation of secondary phosphates has also been advanced as one of three plausible mechanisms for the uptake of trace elements (e.g., REE) by bones during early diagenesis (Trueman and Tuross 2002; Herwartz et al. 2013). To our knowledge, the primary reason that phosphatization has not been advanced (to date) as a mechanism promoting molecular preservation is because the osteocytes, blood vessels, and fibrous matrix fragments recovered from fossil bones have been found to be enriched largely in iron, with minor amounts of Al, K, and Si is some specimens, but with negligible amounts of P in most cases. It is logical and intriguing to think that micro-sclae phosphatization may also contribute to molecular preservation (perhaps via mineral encapsulation; cf. Sykes et al. 1995), especially if oxidizing conditions continue to be found to be favorable, but at present there is no direct evidence of phosphatization occurring (again, to our knowledge). For this reason, we have made no addition or change to the main text regarding this reviewer comment. However, we look forward to further evaluating this possibility in studies of other fossil specimens in the future, and thank the reviewer for this thought-provoking comment.

---

Reviewer 2 Report

Dear authors

I read with interest this study about fossildiagenesis. My concern is related to the taphonomic approch suggested in the title, which in part is trully appliable, but lacks bioestratinomic data as well (and very important) geologic background. Some conclusions are based on an interpreted fluvial tranport downstream, and posterior deposition in an estuary. It can be truth (based on the updated interpretation for the Judith River Formation), but you did not present data to support those interpretations. Please, see the attached file for better understanding my concern.
I think that you should not emphasize taphonomy as the main approach but focus on fossildiagenesis alterations, so, the manuscript would be without potential misinterpretations. In other hand, you could provide a geologic section, illustrating the sedimentary facies and architetural elements to support those intepretation, also presenting bioestratinomic data for those bones. Honestly, I think that focusing on the fossildiagenesis would be more profitable for the proposed study.

Author Response

Reviewer 2, Comment 1: “My concern is related to the taphonomic approch suggested in the title, which in part is trully appliable, but lacks bioestratinomic data as well (and very important) geologic background. Some conclusions are based on an interpreted fluvial tranport downstream, and posterior deposition in an estuary. It can be truth (based on the updated interpretation for the Judith River Formation), but you did not present data to support those interpretations. Please, see the attached file for better understanding my concern.”

Response: We believe this concern expressed by the reviewer is largely one of semantics. Diagenesis, by definition, is all processes affecting a sediment and its fossil content from deposition until metamorphosis (Martin 1999). Diagenesis is considered a subfield of the discipline of taphonomy. Specifically, Behrensmeyer and Kidwell (1985) divided taphonomy into three successive sub-disciplines: necrology, biostratinomy, and diagenesis. Thus, all of our discussions of diagenetic alterations which occurred to the remains of MOR 2598 also simultaneously constitute taphonomic observations and interpretations. Thus, because our manuscript includes extensive considerations of the taphonomic history of MOR 2598, feel that use of the term “Taphonomic” in our title is appropriate.

However, it is true that our discussion of biostratinomic aspects of the taphonomy of MOR 2598 is relatively brief, including in comparison to our “Part I” study of T. rex MOR 1125 published last year in this same Special Issue (Ullmann et al. 2021). This was also noted by another reviewer (to the Editor: please see our response to Reviewer 4 Comment 2 below). Part of our brevity reflects our efforts to write concisely, but the majority of it stems from limited available geologic and biostratinomic information. No other biostratinomic or stratigraphic data were acquired during the collection of MOR 2598 in 2006 and 2007 by the Museum of the Rockies. In fact, all available sedimentologic, stratigraphic, and biostratinomic data are already presented in our Taphonomic and Geologic Context paragraph. Although neither a quarry map nor stratigraphic column were gathered by the MOR crews, this section of our manuscript still presents a wealth of valuable information about the post-mortem history of MOR 2598 which we factor into our interpretations in the Discussion. In short, we have presented all data that we can about the geologic context and biostratinomy of MOR 2598, and we feel that we drew conservative conclusions about its post-mortem history based on these data. Given the absence of a more specific objection by the reviewer to any of our taphonomic interpretations, we have not made any changes to the manuscript based on this comment by the reviewer.

---

Reviewer 2, Comment 2: “I think that you should not emphasize taphonomy as the main approach but focus on fossildiagenesis alterations, so, the manuscript would be without potential misinterpretations. In other hand, you could provide a geologic section, illustrating the sedimentary facies and architetural elements to support those intepretation, also presenting bioestratinomic data for those bones. Honestly, I think that focusing on the fossildiagenesis would be more profitable for the proposed study.”

Response: As discussed in our response to the previous comment, diagenesis is considered a sub-discipline of taphonomy (Behrensmeyer and Kidwell 1985). Thus, all considerations of diagenesis constitute taphonomic considerations as well. Yes we emphasize diagenetic alterations and interpret them, but all of this falls under the realm of taphonomy. Please see our response to the previous comment for further information and details about the geologic data we can (and unfortunately cannot) present in the manuscript.

---

Reviewer 2, Comment 3: Line 3, “Maybe I missed something. Where is or what is the part I?”

Response: Part I was our previous manuscript on T. rex MOR 1125 published last year in this same Special Issue of Biology (Ullmann et al. 2021).

---

Reviewer 2, Comment 4: Line 53, replace “invertebrates and vertebrates” with “animals”

Response:  We have changed this phrase to “plants and animals”, as plants is also needed to fully cover the invertebrate category.

---

Reviewer 2, Comment 5: Line 56, remove “charismatic”

Response: Accepted.

---

Reviewer 2, Comment 6: Line 74, remove “to some”

Response: Rejected. Many paleontologists are convinced by the abundance of data in support of molecular preservation in Mesozoic fossils, while only a few still voice opposition to this possibility, and those who do so advance illegitimate and flawed arguments in their objections (see Schweitzer et al. 2019 for a recent review of these debates). Therefore, we elect to keep the phrase “to some” because it is accurate, because only “some” paleontologists find this topic controversial.

---

Reviewer 2, Comment 7: Line 138, “explain abbreviation (the reader need to be presented to the abbreviation in the first use)”

Response: Accepted.

---

Reviewer 2, Comment 8: Line 170, remove “almost certainly”

Response: Rejected. Removing these two words would make the statement 100% certain, which we are not and cannot be. Thus, “almost” is necessary to leave room for a small degree of uncertainty in our conclusion.

---

Reviewer 2, Comment 9: Line 387, “explain it in the geologic setting section”

Response: As discussed in our response to Reviewer 2 Comment 1, we have presented all available sedimentologic, stratigraphic, and biostratinomic data about MOR 2598 in the Taphonomic and Geologic Context section. Therefore, we have “explained it”, to use the reviewer’s words, as best we can.

---

Reviewer 2, Comment 10: Line 388, “bioestratinomic data can be contrary to this idea! The bone is partially fragmented, so, it has tranport involved and some residence time (sensu Kidwel”

Response: In lines 142-145 we noted how we attribute the incompleteness of the tibia to modern weathering and erosion, not transport and breakage prior to fossilization. Thus, the bone would have originally been complete upon burial, then fragmented millions of years later. This scenario is compatible with the full articulation of the hind limb, which in turn supports rapid burial, as we state in line 388. Thus, we made no changes to the text based on this comment by the reviewer.

---

Reviewer 2, Comment 11: Line 391, “it is not in accordance with ‘rapid burial’ as stated in the previous sentences”

Response: There is no set definition of ‘rapid burial’. It is open to subjective interpretation, and is often used to encompass varied timeframes by different researchers. However, it is generally used when paleontologists see evidence for “exceptional” preservation, such as fully articulated skeletons or portions. Since the hind limb of MOR 2598 was found fully articulated, and would therefore have to have been buried while ligaments and other structural soft tissues still held the skeletal elements together (to the Editor, please also see our response to Reviewer 1 Comment 3), then we feel the phrase ‘rapid burial’ is appropriate for MOR 2598. Additionally, ligaments are known to persist up to months or even a few years in carcasses exposed in modern temperate environments (Hill and Behrensmeyer 1984), so our use of the term “rapid burial” for a timeframe of “within a few years postmortem” (in line 391) is justified.

---

Reviewer 2, Comment 12: Line 393, “why brief if you have no presented data about bioestratinomy”

Response: As noted in our response to Reviewer 2’s Comment 1 above, we present a suite of biostratinomic data about MOR 2598 in the Taphonomic and Geologic Context section (e.g., articulation state, entombing lithology, etc.). Second, we interpret “brief subaerial decay” here because if the carcass were exposed for more than a few years then decay would advance far enough for the ligaments to break down and the bones to become disarticulated upon burial (cf. Hill and Behrensmeyer 1984). Since that did not happen to MOR 2598, then its carcass must have been subaerially exposed for < ~5 years, so use of the adjective “brief” for this timespan seems appropriate.

---

Reviewer 2, Comment 13: Line 394, “again, how could it be inferred if you do not have presented biostratinomic data? Additionally, carcaces can floatted and be transported several kilometers without damage to the skeletal parts. Caution in those interpretations, your data are molecular, paleodepositional setting and biostratinomic aspects can not be directly inferred from them”

Response: Again, please see our response to Reviewer 2 Comment 1 regarding the suite of biostratinomic data we present about MOR 2598 in the Taphonomic and Geologic Context section. Regarding our interpretations in line 394, we believe we have exercised “caution” as stated by the reviewer because we used the word “probable” in inferring that the hind limb was transported along the fluvial channel. “Probable” implies an appropriate degree of uncertainty based on the limited available data and the fact that we were not there to directly observe what happened.

---

Reviewer 2, Comment 14: Line 395, “this study also do not explain how they know that it was a channel of an estuary, or why it is fluvial in its origin”

Response: We acknowledge that Schweitzer et al. (2009) (our Ref #30) did not adequately explain why they interpreted the burial site as a fluvial channel, but it was based upon: 1) prior summaries of the Judith River Formation as representing largely fluvial deposits and; 2) the sedimentology of the entombing strata, namely a thick sequence of trough cross-stratified sandstones. We had already cited two prior studies (Rogers and Kidwell 2000; Rogers et al. 2016) in line 406 regarding the general lowland fluvial system-nature of Judith River Formation deposits, but to clarify the second reason listed above we have added the detail about the “trough cross-bedded” character of the entombing sandstones to line 140 of the Taphonomic and Geologic Context section.

---

Reviewer 2, Comment 15: Line 397, “no data to support it! (at least, no data presented here or in the cited literature)”

Response: We acknowledge that our interpretation of disarticulation of the leg from the rest of the body having occurred due to fluvial currents is “likely” in this line, not certain. It is the most parsimonious interpretation based on the sedimentology and stratigraphy of the sediments exposed in the MOR 2598 quarry (also see our response to Comment 14 by this reviewer for further geologic details), and it is a hypothesis that is compatible with the general fluviodeltaic character of Judith River ecosystems in general (Rogers and Kidwell 2000; Rogers et al. 2016). We therefore have made no changes to the text based on this comment by the reviewer.

---

Reviewer 2, Comment 16: Line 402-403, “It is crucial to illustrated that it is a sinlge channel instead amalgamated channels. Again, those interpretations are not supported by your data”

Response: We do not agree with the reviewer that it is “crucial” to show this, as the two alternatives have no meaningful influence on our ultimate interpretations for the geochemical/diagenetic history of MOR 2598 primarily examined in this manuscript. Moreover, our sentence in the marked lines interprets that burial took place in a “well-established lowland channel rather than a recently-formed avulsion channel” because the latter would stratigraphically form a single, thin, lens-shaped channel deposit, whereas the former would produce a stacked succession of cross-stratified sandstone layers (Reineck and Singh 1980). The strata exposed in the quarry clearly match the latter scenario, supporting our interpretation of the channel having existed in this area for quite some time prior to the burial of MOR 2598 within the channel.

---

Reviewer 2, Comment 17: Line 406, “Those studies are from your area? If so, cite them before (geologic context). If not, I keep my concern about those depositional interpretations (I checked Roger's paper and it is not from your area)”

Response: We disagree. Rogers’ stratigraphic studies (Rogers and Kidwell 2000; Rogers et al. 2016) were conducted in the same region: north central Montana, specifically near the Missouri Breaks region. Moreover, their paleoenvironmental interpretations were based on multiple stratigraphic columns across the region of central Montana, and thus characterize the depositional settings of the Judith River Formation as a whole, which includes where MOR 2598 was buried within Judith River deposits (near Malta, Montana). Based on this, we have accepted the reviewer’s suggestion to cite Rogers and Kidwell (2000) and Rogers et al. (2016) earlier in the manuscript, and have added a new sentence to the Taphonomic and Geologic Context section to do so. It reads as follows: “Schweitzer et al. (2009) concluded that these strata were deposited in a fluvial channel within the overall lowland fluviodeltaic system of the Judith River ecosystems (Rogers and Kidwell 2000; Rogers et al. 2016).”

---

Reviewer 2, Comment 18: Line 407-413, “Again, as reader I only see interpretations. No data”

Response: Please see our responses to Comments 1, 10-13, 15, and 16 by Reviewer 2, which explain how we presented sedimentologic, stratigraphic, and biostratinomic data in the manuscript and conservatively interpreted them based on logic and parsimony.

---

Reviewer 2, Comment 19: Figure 7, “You had not present data for that part of the diagram”

Response: As explained in our response to Reviewer 2’s Comment 18 above, we have inferred that these universal (and largely unavoidable) events took place based on the preservation state of the specimen, and all of these inferences are based on logic, modern actualistic studies (e.g., Hill and Behrensmeyer 1984; Trueman et al. 2004; Daniel and Chin 2010; Fernandez-Jalvo et al. 2010), and parsimony. For these reasons, we feel that the upper two rows of this figure marked by the reviewer portray sensible taphonomic and diagenetic events that we can reconstruct with confidence, and therefore we have elected to keep these rows in the figure without any changes.

---

Reviewer 2, Comment 20: Line 674, “See bioestratinomic concerns above”

Response: Please see our response to Reviewer 2’s Comments 10, 11, and 12 above, which explain how and why we concluded that MOR 2598 was preserved via rapid burial.

---

Reviewer 3 Report

Congratulations, this is an excellent contribution on a very interesting topic, not only for paleontologists but for geochemists and many others in the Earth sciences. I am sure this will be a highly citated paper once it gets published.

Author Response

Reviewer 3, Comment 1: “Congratulations, this is an excellent contribution on a very interesting topic, not only for paleontologists but for geochemists and many others in the Earth sciences. I am sure this will be a highly citated paper once it gets published.”

Response: Thank you! We are grateful for your kind comments. We made no changes based on this reviewer’s single comment.

---

Reviewer 4 Report

Comments on the manuscript entitled “Taphonomic and diagenetic pathways to protein preservation,  part II: the case of Brachylophosaurus canadensis specimen  MOR 2598" submitted by Ullmann et al.

This paper is similar to a previous one: “Taphonomic and Diagenetic Pathways to Protein Preservation, Part I: The Case of Tyrannosaurus rex Specimen MOR 1125".
In both papers, the authors claim that the selected bones are macroscopically well preserved, although the bones are brown, almost black.  Taphonomic analyses are succinct (5 lines), not illustrated.
Is the bone still in apatite? Is it enriched in some chemical elements? What about the crystallinity? Is the bone histology preserved? The fragment used for the analyses could have been observed with a SEM microscope.       

Composition is reduced to REE of bones. What about the composition of embedding sediment? It has been shown that REE are useful to decipher the geological story of bone, but they are not enough, although a long and detailed discussion. Several independent techniques are usually done and the results are compared and discussed.

Despite the fact that the manuscript is dedicated to the preservation vs diagenesis of a protein, nothing is known about the quantity of organic matter. Similarly, data about the composition of the organic matter is not known, despite several non destructive techniques exist, so that it is possible to differentiate lipids, proteins....

Collagen in fossils collected in Jurassic sediments is known since the pioneering works of Wyckoff et al. using TEM and aminoacid analysis.

Author Response

Reviewer 4, Comment 1: “This paper is similar to a previous one: “Taphonomic and Diagenetic Pathways to Protein Preservation, Part I: The Case of Tyrannosaurus rex Specimen MOR 1125". In both papers, the authors claim that the selected bones are macroscopically well preserved, although the bones are brown, almost black.”

Response: This manuscript is indeed intentionally very similar to our previous “Part I” published last year in this same Special Issue of Biology (Ullmann et al. 2021). We submitted our complimentary research on the specimen examined herein, Brachylophosaurus MOR 2598, in response to an enthusiastic request from Editor Peter Gao, who expressed excitement about our “Part II”. To clarify the point about each specimen being “macroscopically well preserved”, in paleontology bones are considered to be well preserved if their three-dimensional anatomy is intact. Neither color, chemistry, nor mineralogy matter when a paleontologist uses this phrase: it is simply a quick assessment in the general sense, gauging whether the form of the bone appears more or less like it should have when the animal was alive. Both the T. rex femur examined by Ullmann et al. (2021) and the tibia of Brachylophosaurus MOR 2598 examined herein are visually “well preserved” in that they retain their 3D shape very well. Additionally, the change in color of fossil bones to brown is extremely common due to uptake of fluorine from natural waters (Elorza et al. 1999), and it merely demonstrates that microscopic recrystallization from hydroxyapatite to fluorapatite occurred – again, a nearly-universal occurrence in the preservation of bones as fossils (Elorza et al. 1999). Thus, in short, because of the commonality of this minor chemical alteration, paleontologists consider specimens “well preserved” based solely on shape. We believe our image in Figure 1B sufficiently shows how, though fractured, the bone retains its original shape, thus showing it is “macroscopically well preserved”; we therefore have made no changes to the text based on this comment.

---

Reviewer 4, Comment 2: “Taphonomic analyses are succinct (5 lines), not illustrated.”

Response: We admit that our discussion of ‘traditional’ aspects of the taphonomy of MOR 2598 is relatively brief, including in comparison to our “Part I” study of T. rex MOR 1125 published last year in this same Special Issue (Ullmann et al. 2021). Part of this reflects our efforts to write concisely, but the majority of it stems from limited available information. In fact, it is not possible for us to provide further information at this time because no other taphonomic or stratigraphic data were acquired during the collection of MOR 2598 in 2006 and 2007 by the Museum of the Rockies. All available sedimentologic, stratigraphic, and ‘traditional’ taphonomic data are already presented in our Taphonomic and Geologic Context paragraph. Please note that this section constitutes 20 lines of text, not 5 as stated by the reviewer, and that it presents information on all of the following: 1) the portion of the skeleton that was recovered [left hind limb]; 2) the articulation state of its bones upon discovery [articulated]; 3) the ontogenetic age of the individual [subadult]; 4) its geologic age [Campanian]; 5) the geologic Formation the bones were recovered from [Judith River Fm.]; 6) the type of sediment they were collected from [sandstone] and the inferred paleoenvironment of burial [fluvial channel]; 7) the completeness of the tibia examined herein [incomplete, distal end missing]; 8) the color of the bones [light to dark brown]; 9) the degree of weathering [none]; 10) the degree of abrasion [none]; 11) the degree of fracturing [extensive] of the tibia; 12) the types of bone fractures primarily observed [transverse and longitudinal] and their likely origin [post-fossilization compaction], and; 13) the state of permineralization [none]. Thus, although neither a quarry map nor stratigraphic column can be presented as visual figures, this section of the manuscript still presents a wealth of valuable information about the post-mortem history of MOR 2598 which we factor into our taphonomic interpretations in the Discussion. Also, Figure 1B presents a photograph of the tibia we examined, and this image visually shows seven of the taphonomic aspects listed above (i.e., how it is well-preserved morphologically, lacks signs of weathering or abrasion, is incomplete, brown in color, and extensively fractured with transverse and longitudinal breaks). Thus, we feel that Figure 1B already provides the best taphonomic visual that we can offer at this time, and that the Taphonomic and Geologic Context section already presents everything that we can regarding ‘traditional’ taphonomic data about MOR 2598. For these reasons, we have not made any changes or additions to the figures or manuscript text based on this comment by the reviewer.

---

Reviewer 4, Comment 3: “Is the bone still in apatite? Is it enriched in some chemical elements? What about the crystallinity? Is the bone histology preserved?”

Response: We will answer these four questions one at a time: For the first question, while we are 99.999% certain that the bone is preserved as fluorapatite based on its visual appearance, color, and the nearly-universal nature of this minor mineralogical transition during the fossilization of bones and teeth (Keenan 2016), x-ray diffraction (XRD) was not performed to verify this. Although we could reach out to a new collaborator to partner with in order to conduct XRD, that would take time (likely weeks in order to fit the instrument type into their existing schedule). We feel that because this is such a minor point, it is not worth delaying the publication of our manuscript to confirm this near-certain prediction. However, to acknowledge that this conclusion about the specimen’s mineralogy is an interpretation, we have added the following sentence to the Taphonomic and Geologic Context section of the text: “All of the skeletal elements are brown in color (e.g., Figure 1B), indicating their mineralogy has likely been transformed from hydroxyapatite to fluorapatite, which is typical of bone fossilization (Elorza et al. 1999; Keenan 2016).”

For the second question, we are not sure what the reviewer’s question of “Is it enriched in some chemical elements?” is intending to ask. Our manuscript discusses elemental enrichment (or lack thereof) throughout, such as clear enrichment in U and HREE but minimal uptake of Y (see Figure 2 and the corresponding section of the Results). We therefore think any questions about elemental enrichment should already be well covered by our manuscript, and have made no additions to the text based on this question by the reviewer.

For the third question, crystallinity is often identified by XRD. Though we did not perform XRD, crystallinity of bones is almost universally found to have increased during fossilization due to recrystallization (Person et al. 1996; Trueman et al. 2008), so our specimen would likely also show elevated crystallinity. Although we could work with a collaborator to perform XRD to confirm this (by identifying the specific crystallinity index [CI] of the tibia of MOR 2598), we see no clear benefit that could be acquired by taking the time to do this because CI values have been found to exhibit poor correlation with the degree of chemical alteration of fossil bones (Pucéat et al. 2004). In their words, Pucéat et al. (2004, p. 83) found that “crystallinity index is a poor criterion for determining if a sample has been altered since deposition” as “strong geochemical perturbations may occur without detectable recrystallization”. Thus, we feel that assessment of the CI of MOR 2598 would not yield meaningful results, and that it is not worth examining this somewhat-tangential question at this time.

For the fourth question, a short answer is “yes”. Schweitzer et al. (2016) examined the histology of the femur of MOR 2598 and found it to exhibit excellent microstructural preservation. They presented a figure of its histology (their figure 10B). We cited Schweitzer et al. (2016) for this fact in line 390 (near the beginning of our Discussion). To clarify that they also figured its histology, we have accepted another reviewer’s suggestion to add a note specifying the reference to their figure 10B. This citation is now formatted as follows: “(figure 10B of [70])”.

---

Reviewer 4, Comment 4: “The fragment used for the analyses could have been observed with a SEM microscope.”

Response: We are uncertain if the reviewer means that the ‘raw’ bone fragment (prior to its thick sectioning for trace element analyses) or the thick section of it used in our LA-ICPMS analyses could be observed under SEM. However, we do not see any major insight that could be gained from analyzing either of these samples under SEM. The thick section has been modified by cutting and polishing, and observation of the original bone fragment it was cut from would merely reveal the style of breaks created by breaking the fragment off the rest of the bone. Thus, we see no meaningful new information that could be garnered by SEM imaging either of these samples. Also, in terms of chemistry, the bone is brown in color (see our Figure 1B and lines 143-144 of our text), indicating it was converted from hydroxyapatite into fluorapatite, which represents a minor mineralogical alteration via incorporation of fluorine into the crystal structure that is known to be typical of bone fossilization (Elorza et al. 1999; Keenan 2016). Therefore, it is likely that energy dispersive spectroscopy (EDS) coupled with SEM would merely identify primarily Ca and P, as expected in both of these apatite minerals (Keenan 2016). Thus, we see no relevant need for samples to be analyzed by SEM, and have not done so.

---

Reviewer 4, Comment 5: “Composition is reduced to REE of bones. What about the composition of embedding sediment? It has been shown that REE are useful to decipher the geological story of bone, but they are not enough, although a long and detailed discussion. Several independent techniques are usually done and the results are compared and discussed.”

Response: First, we examine more than just the REE elements (La-Lu). As shown in our Table 1 and discussed throughout the manuscript, we also characterized and discussed the importance of the fossil’s content of other trace elements, namely U, Sc, Sr, Y, Ba, Mn, Fe, and Th. Concentration profiles of many of these elements were presented in Figure 2 and provided helpful information about the diagenetic history of MOR 2598, which we interpreted in the Discussion. Therefore, we believe that the reviewer’s statement does not appropriately summarize the analyses we conducted.

Second, we are aware that useful information (e.g., trace element supply and sources in the local area) can be gathered from similarly analyzing associated sediment samples (e.g., Kowal-Linka and Jochum 2015), but unfortunately no sediment sample was available for this study. The bones of MOR 2598 were prepared years ago, in which time all sediment was removed from the bones of the specimen and discarded. While this is unfortunate, numerous studies (e.g., Trueman 2007; Trueman et al. 2008, 2011; Grandstaff and Terry 2009; Koenig et al. 2009; Herwartz et al. 2011, 2013; Kowal-Linka et al. 2014; McCormack et al. 2015; Zigaite et al. 2016; Decree et al. 2018; Suarez et al. 2019; Wang et al. 2020; Bosio et al. 2021; Ferrante et al. 2021; Titus et al. 2021; Ullmann et al. 2020, 2021) have demonstrated that analyses of trace element signatures in fossil bones alone can yield valuable, informative insights into the diagenetic histories that they have experienced, even if the entombing sediments are not similarly analyzed – indeed, none of the studies cited above examined the associated sediments. Therefore, there is an established precedent that paired analyses of associated sediment samples are not required in order to draw meaningful insights. For this reason, we do not believe that lack of analyses of the entombing sediment is a serious issue, and are confident that our study, like that of Ullmann et al. (2021) and the numerous others cited above, provides valuable insights that should be shared with the paleontologic community via publication.

Finally, while it is true than “several independent techniques” could be performed to further examine the diagenetic history of MOR 2598, it is unclear why they are needed and which the reviewer is suggesting might be helpful. The vast majority of the studies cited just above performed only solution-ICPMS or laser LA-ICPMS, and did not conduct other analytical techniques because ICPMS results provide a vast wealth of information on their own. Thus, we feel that our LA-ICPMS data are sufficient to address the questions we sought to answer, and have not conducted additional methods at this time.

---

Reviewer 4, Comment 6: “Despite the fact that the manuscript is dedicated to the preservation vs diagenesis of a protein, nothing is known about the quantity of organic matter. Similarly, data about the composition of the organic matter is not known, despite several non destructive techniques exist, so that it is possible to differentiate lipids, proteins.... Collagen in fossils collected in Jurassic sediments is known since the pioneering works of Wyckoff et al. using TEM and aminoacid analysis.”

Response: The reviewer is correct that collagen has been identified previously in numerous other fossils of varied geologic ages. Since we cite numerous prior studies which identified collagen in fossils in the second paragraph of our Introduction (specifically Refs 18-22 and 24-47), we feel this point is well established within our text already. Regarding chemical data on the composition of biomolecules recovered from MOR 2598, Schweitzer et al. (2009) performed ELISA analyses on protein extracts from the femur of MOR 2598, and identified a positive signal for both collagen I and osteocalcin. Their results (their figure S3a) identified the fossil bone to retain ~25% of the amount of collagen that should be present in modern cortical bone tissues – a reduced amount as should be expected after prolonged decay. They also performed two other immunoassays (Western blot, in situ immunohistochemistry) which additionally found evidence for the presence of the protein collagen I in this specimen, as well as LC/MS/MS analyses which recovered peptides of collagen I from the femur. Our study examines the physical and geochemical taphonomy of this specimen, in an attempt to explain how the molecular signatures of collagen I and other proteins identified by Schweitzer et al. (2009) in this specimen were able to persist for tens of millions of years. Thus, in short, chemical data supporting the preservation of collagen in this specimen were previously acquired by Schweitzer et al. (2009), which we cite frequently throughout our manuscript, and we see no need to further replicate those analyses here because they are tangential to the primary (taphonomic) research questions being addressed in our manuscript.

---

Round 2

Reviewer 4 Report

Taphonomy and diagenesis modifications are various and complex, so that to decipher the geological story of a sample is difficult. To take into account only one paramter is not enough.

However, papers dealing with REE contents are rare, so despite its lacunae, the manuscript deserves to be published.